# BQ-NCO: Bisimulation Quotienting for Efficient Neural Combinatorial Optimization

**Darko Drakulic, Sofia Michel, Florian Mai,[*] Arnaud Sors, Jean-Marc Andreoli**
Naver Labs Europe
`firstname.lastname@naverlabs.com`

## Abstract

Despite the success of neural-based combinatorial optimization methods for end-to-end heuristic learning, out-of-distribution generalization remains a challenge. In this paper, we present a novel formulation of Combinatorial Optimization Problems (COPs) as Markov Decision Processes (MDPs) that effectively leverages common symmetries of COPs to improve out-of-distribution robustness. Starting from a direct MDP formulation of a constructive method, we introduce a generic way to reduce the state space, based on Bisimulation Quotienting (BQ) in MDPs. Then, for COPs with a recursive nature, we specialize the bisimulation and show how the reduced state exploits the symmetries of these problems and facilitates MDP solving. Our approach is principled and we prove that an optimal policy for the proposed BQ-MDP actually solves the associated COPs. We illustrate our approach on five classical problems: the Euclidean and Asymmetric Traveling Salesman, Capacitated Vehicle Routing, Orienteering and Knapsack Problems. Furthermore, for each problem, we introduce a simple attention-based policy network for the BQ-MDPs, which we train by imitation of (near) optimal solutions of small instances from a single distribution. We obtain new state-of-the-art results for the five COPs on both synthetic and realistic benchmarks. Notably, in contrast to most existing neural approaches, our learned policies show excellent generalization performance to much larger instances than seen during training, without any additional search procedure. Our code is available at: url.

## 1 Introduction

Combinatorial Optimization Problems (COPs) are crucial in many application domains such as transportation, energy, logistics, etc. Because they are generally NP-hard [12], their resolution at real-life scales is mainly done by problem-specific heuristics, which heavily rely on expert knowledge. Neural combinatorial optimization (NCO) is a relatively recent line of research that focuses on using deep neural networks to learn such heuristics from data, possibly exploiting regularities in problem instances of interest [5, 9]. Among NCO methods, the so-called constructive approaches view the process of building a solution incrementally as a sequential decision making problem, which can naturally be modeled with Markov Decision Processes (MDPs). Although many previous works have successfully used this strategy, the process of formulating an appropriate MDP (esp. the state and action spaces) is usually specific to each problem (e.g. [28] for routing problems, [49] for jobshop scheduling and [20] for bin packing). These design choices have a considerable impact when solving the MDP. In particular, exploiting the COP's symmetries can boost the efficiency and generalization of neural solvers (e.g. [29, 24] leverage the symmetries of routing problems in Euclidean graphs).

In this paper, we present a generic and principled framework to cast any COP as an MDP and then a way to account for common COPs' symmetries to design a more efficient MDP. More precisely,

[*]IDIAP Research Institute, Work done during an internship at Naver Labs Europe

37th Conference on Neural Information Processing Systems (NeurIPS 2023).

given a user-defined *solution space*, we show how to automatically derive a *direct MDP*, where the states are partial solutions and the actions are the construction steps. While the application of this framework to some COPs encompasses some previously proposed MDPs (e.g [28], for routing problems), to the best of our knowledge, we are the first to provide a way to derive an MDP for any COP and prove the equivalence between the optimal MDP policies and solving the COP.

Next, noting the limitations of using the partial solutions as states, we introduce a generic *bisimulation* mapping that allows to reduce the state space. In particular, we show that for problems that satisfy a *recursion property*, the bisimulation simply maps a partial solution to a new (induced) instance, which corresponds to the remaining subproblem when the partial solution is fixed. As many partial solutions can induce the same instance, the resulting bisimulation quotiented (BQ-)MDP has a significantly smaller state space. Also, it enables more efficient learning by avoiding to independently learn the policy at states which are equivalent for the COP at hand. In contrast to previous works ([29, 24]), the symmetries that we exploit are not linked to the Euclidean nature of the graphs but to the recursive property of the problems, which is very common in CO as it includes the Optimality Principle of Dynamic Programming [3, 6].

We illustrate our framework on five well-known COPs: the Euclidean and Asymmetric Traveling Salesman Problems (TSP, ATSP), the Capacitated Vehicle Routing Problem (CVRP), the Orienteering Problem (OP) and the Knapsack Problem (KP). Furthermore, we propose a simple transformer-based architecture [43] for these problems, well-suited to the BQ-MDPs, and requiring only minor variations to work for the five problems. For each problem, we train our policy by imitation of expert trajectories derived from (near) optimal solutions of small instances sampled from a single distribution. We test on both synthetic and realistic benchmarks of varying size and node distributions. Our model provides new state-of-the-art results on the five problems and exhibits excellent out-of-distribution generalization, especially to larger instances, a well-recognized challenge for NCO methods [22, 32]. Notably with a single greedy rollout, our learned policies outperform state-of-the-art end-to-end learning-based approaches for instances with more than 200 nodes. We show that we can further improve the performance with a beam-search (using more computation) while we can significantly speed-up the execution (at the cost of a slight performance drop) by replacing the quadratic transformer model by a linear attention-based model, the PerceiverIO [19].

In summary, our contributions are as follows: 1) We present a generic and principled framework to derive a direct MDP given any COP with minimal requirements; 2) We propose a method to reduce the direct MDPs via symmetry-focused bisimulation quotienting and define an explicit bisimulation for the class of recursive COPs; 3) We design an adequate transformer-based architecture for the BQ-MDPs, with only minor adaptations to work for the TSP (both Euclidean and asymmetric versions), CVRP, OP and KP; 4) We achieve state-of-the-art generalization performance on these five problems, significantly out-performing other neural-based constructive methods.

## 2 Combinatorial Optimization as a Markov Decision Problem

In this section, we propose a generic framework to represent any Combinatorial Optimization Problem (COP) as a Markov Decision Process (MDP) amenable to standard Machine Learning techniques. We denote a COP instance by:

$$\min_{x \in X} f(x),$$

where $X$ is the finite, non-empty set of *feasible solutions*, while the *objective* $f$ is a real-valued function whose domain contains $X$. The complexity of CO is due to the cardinality of $X$, which, although finite, is generally exponential in the problem size. Constructive approaches to CO build a solution sequentially by growing a partial solution at each step. A required assumption of constructive heuristics (although often left implicit) is that the feasibility of the final solution can be ensured through tractable conditions on the partial solutions at each step of the construction process.

### 2.1 Solution Space

Let us denote by $\mathcal{X}$ the set of all possible *partial solutions* which can be obtained by a given construction procedure. We assume that $\mathcal{X}$ is equipped with an operation $\circ$ having a neutral element $\epsilon$. Informally, $\epsilon$ is the "empty" partial solution and if $x, y$ are partial solutions, then $x \circ y$ denotes the result of applying the sequence of construction steps yielding $x$ followed by that yielding $y$. We

denote by $\mathcal{Z}\subset\mathcal{X}$ the subset of partial solutions obtained from $\epsilon$ by just one construction step. By identification, we call the elements of $\mathcal{Z}$ *steps* and assume that any partial (a fortiori feasible) solution can be obtained as the composition of a sequence of steps. We can now define the structure of the solution space as required by our framework.

**Definition 1** (Solution Space)**.** *A* solution space *is a tuple* $(\mathcal{X}, \circ, \epsilon, \mathcal{Z})$ *where* $(\mathcal{X}, \circ, \epsilon)$ *forms a monoid (see Appendix G.1 for background), and the step set* $\mathcal{Z}\subset\mathcal{X}\backslash\{\epsilon\}$ *is a generator of* $\mathcal{X}$*, such that any element of* $\mathcal{X}$ *has a finite positive number of step decompositions:*

$$\forall x \in \mathcal{X},\ 0 < |\{z_{1:n} \in \mathcal{Z}^n :\ x = z_1 \circ \cdots \circ z_n\}| < \infty. \tag{1}$$

Thus, solving a COP instance $(f, X)$ given a solution space $\mathcal{X}$ containing $X$ amounts to finding a sequence of steps $z_{1:n}$ such that $z_1\circ\cdots\circ z_n\in \arg\min_{x\in X} f(x)$, a task which can naturally be modeled as an MDP, as shown below.

**Examples of Solution Spaces.** We provide examples of solution spaces for two classical COPs. In the Euclidean Traveling Salesman Problem (TSP), an instance is defined by a set of nodes (including a depot) associated with points in a Euclidean space and the goal is to find the shortest tour that starts and ends at the depot and visits each other node exactly once. A partial solution for the TSP can be represented by a sequence of nodes. Therefore we can define $\mathcal{X}_{\text{TSP}}$ as the set of finite sequences of nodes. In the Knapsack Problem (KP), an instance is defined by a set of items associated with weight and value features, and the goal is to select a subset of these items such that the sum of their weights does not exceed a given capacity while their cumulated value is maximized. We can define $\mathcal{X}_{\text{KP}}$ as the set of finite sets of items. For TSP (resp. KP), the *operator* $\circ$ is sequence concatenation (resp. set disjoint union), the *neutral element* $\epsilon$ is the empty sequence (resp. set) and a *step* is a sequence (resp. set) of length 1. Note that these problems admit other (e.g. graph-based rather than set or sequence-based) solution spaces. In fact, a solution space is not intrinsic to a problem, nor vice-versa.

### 2.2 The Direct MDP

Given a solution space $(\mathcal{X}, \circ, \epsilon, \mathcal{Z})$, we consider the set $\mathcal{F}_{\mathcal{X}}$ of instances $(f, X)$ where the set of feasible solutions $X$ is a finite, non-empty subset of $\mathcal{X}$, and the objective function $f\in\mathbb{R}^{\mathcal{X}}$ is defined on $\mathcal{X}$ rather than just $X$, *i.e. it is well-defined for any partial solution*. Now, given an instance $(f, X)\in\mathcal{F}_{\mathcal{X}}$, we can derive its *direct* MDP, denoted $\mathcal{M}_{(f,X)}$, as follows. Its **state space** is the set $\bar{X}=\{x\in\mathcal{X}\colon \exists y\in\mathcal{X},\ x\circ y\in X\}$ of partial solutions which can potentially be expanded into a feasible one. Its **action space** is $\mathcal{Z}\cup\{\epsilon\}$, i.e. an action is either a step or the neutral action. Its **transitions** are deterministic, and can be represented as a labeled transition system with the following two rules, where the label of each transition consists of its action-reward pair, placed respectively above and below the transition arrow:

$$x \xrightarrow[f(x)-f(x\circ z)]{z} x\circ z \ \text{ if }\ x\circ z \in \bar{X} \qquad \text{and} \qquad x \xrightarrow[0]{\epsilon} x \ \text{ if }\ x \in X. \tag{2}$$

Here, $x\in\bar{X}$ is a state and $z\in\mathcal{Z}$ is a step. The condition in each rule determines whether the action is allowed. Thanks to the structure of the solution space captured by Def. 1, the direct MDP of an instance $(f, X)$ has three key properties, proved in Appendix F.1. (1) From any state, the number of allowed actions is finite (even if $\mathcal{Z}$ is infinite), therefore $\mathcal{M}_{(f,X)}$ belongs to the simpler class of discrete-action MDPs. (2) From any state, there is always at least one allowed action (i.e. there are no dead-end states). This assumes that one can guarantee if a partial solution can be expanded into a feasible one, as required of the valid states in $\bar{X}$. This is a common assumption of constructive heuristics (often left implicit) and it avoids the complications associated with dead ends in MDPs [27]. (3) In any infinite trajectory in $\mathcal{M}_{(f,X)}$, the number of transitions involving a *step* action is finite, while all the other transitions involve the *neutral* action. Since the neutral action yields a null reward, this means that the return of a trajectory is well defined, without having to discount the rewards. Also, if $a_{1:\infty}$ is the (infinite) sequence of actions of the trajectory, since all but finitely many of them are neutral actions, their composition $a_1\circ a_2\circ\cdots$ is well defined and is called the *outcome* of the trajectory. We can now establish the main result of this section, proved in Appendix F.2:

**Proposition 1** (Soundness of the Direct MDP)**.** *Given a solution space* $\mathcal{X}$ *and an instance* $(f, X)\in\mathcal{F}_{\mathcal{X}}$*, let* $\mathcal{M}_{(f,X)}$ *be its direct MDP. The set* $\arg\min_{x\in X} f(x)$ *is exactly the set of* $x$ *such that there exists an optimal policy* $\pi$ *for* $\mathcal{M}_{(f,X)}$ *where* $x$ *is the outcome of a trajectory starting at* $\epsilon$ *under policy* $\pi$*.*

This result states the exact correspondence between the optimal solutions of a COP instance and the optimal policies of its direct MDP. Thus, the vast corpus of techniques developed to search for optimal policies in MDPs in general is applicable to solving COPs. The direct MDP encompasses many previously proposed MDPs, where the state is a partial solution and each action consists of adding an element to the partial solution, e.g. [23] for graph problems and [7] for the TSP.

## 3 Bisimulation Quotienting

### 3.1 State information and symmetries

Let $(\mathcal{X}, \circ, \epsilon, \mathcal{Z})$ be a solution space and $\mathcal{M}_{(f,X)}$ the direct MDP of an instance $(f, X){\in}\mathcal{F}_{\mathcal{X}}$. In a trajectory of $\mathcal{M}_{(f,X)}$, observe that each non neutral action $z$ "grows" the state from $x$ to $x{\circ}z$, as defined by (2). For many COPs, this is counter-intuitive, because it implies that the state information increases while the number of allowed actions generally decreases. For example, in the TSP, the state contains the sequence of visited nodes, which grows at each step, while the allowed actions are the set of unvisited nodes, which shrinks at each step. At the end, the state may carry the most information (a full-grown feasible solution), but it is not used in any decision, since there is a single allowed action anyway (the neutral action). To address this mismatch between the information carried by the state and the complexity of the decision it supports, we seek to define a new state which captures only the information needed for the continuation of the construction process. To do so, we observe that a partial solution $y{\in}\mathcal{X}$ can be seen both as an *operator on partial solutions* $x{\mapsto}x{\circ}y$ which *grows* its operand, or, alternatively, as the following *operator on instances*, which *reduces* its operand:

$$(f, X){\mapsto}(f * y, X * y) \quad \text{with:} \quad (f * y)(x) = f(y \circ x) \quad \text{and} \quad X * y = \{x | y \circ x \in X\}.$$

In fact, $(f{*}y, X{*}y)$ is the so-called the *tail subproblem* [6] of instance $(f, X)$ after partial solution $y$ has been constructed. We observe that for a given tail subproblem $(f', X'){\in}\mathcal{F}_{\mathcal{X}}$, there can be many pairs of an instance $(f, X){\in}\mathcal{F}_{\mathcal{X}}$ and partial solution $x{\in}\bar{X}$ such that $(f{*}x, X{*}x){=}(f', X')$. For example in the TSP, the tail subproblem consists of finding the shortest path from the last node $e$ of the partial solution back to the depot $o$ of the instance, which visits the set $I$ of not-yet-visited nodes. Therefore we can see that any TSP instance with the same depot $o$ and any node sequence ending at $e$ with not-yet-visited nodes $I$ will lead to the same tail subproblem. This is a strong *symmetry* with respect to the MDP policy: intuitively it means that an optimal policy should produce the same action for these *infinitely* many (instance–partial solution) pairs. Treating them as distinct states, as in the direct MDP, forces a training procedure to learn the underlying symmetry in order to map them into a common representation.

### 3.2 Bisimulation Quotiented MDP

In order to leverage the above "reduction operator" view and its potential symmetry, we define the following "reduced" MDP $\mathcal{M}$. Its **state space** is the set $\mathcal{F}_{\mathcal{X}}$ of instances. Its **action space** is $\mathcal{Z}{\cup}\{\epsilon\}$, as in the direct MDP. Its **transitions** are also deterministic, and can be expressed with two rules, dual of those in (2):

$$(f, X) \xrightarrow[f(\epsilon)-f(z)]{z} (f * z, X * z) \text{ if } X * z \neq \emptyset \quad \text{and} \quad (f, X) \xrightarrow[0]{\epsilon} (f, X) \text{ if } \epsilon \in X. \qquad (3)$$

This MDP is defined at the level of the whole solution space rather than an individual instance. Now, the mapping $\mathbf{\Phi}_{(f,X)}{:}\bar{X}{\mapsto}\mathcal{F}_{\mathcal{X}}$ from the direct to the reduced states defined by $\mathbf{\Phi}_{(f,X)}(x){=}(f{*}x, X{*}x)$ for all $x{\in}\bar{X}$ is a *bisimulation* between $\mathcal{M}_{(f,X)}$ and $\mathcal{M}$ (background in Appendix G.2 and a proof in Appendix F.3). Formally, $\mathcal{M}$ is isomorphic to the quotient of the direct MDP by the bisimulation (precise statement and proof in Appendix F.3), hence is called the Bisimulation Quotiented (BQ-)MDP. By the bisimulation property, for any direct state $x$, the action-reward sequences spawned from $x$ in $\mathcal{M}_{(f,X)}$ and from $\mathbf{\Phi}_{(f,X)}(x)$ in $\mathcal{M}$ are identical. Therefore, there is a one-to-one correspondence between the trajectories of $\mathcal{M}_{(f,X)}$ starting at $\epsilon$ and those of $\mathcal{M}$ starting at $\mathbf{\Phi}_{(f,X)}(\epsilon){=}(f, X)$. Hence, the analogue of Prop. 1 holds for the BQ-MDP:

**Proposition 2** (Soundness of the BQ-MDP). *Let $\mathcal{M}$ be the BQ-MDP of a solution space $\mathcal{X}$, and $(f, X){\in}\mathcal{F}_{\mathcal{X}}$ an instance. The set $\arg\min_{x \in X} f(x)$ is exactly the set of $x$ such that there exists an optimal policy $\pi$ for $\mathcal{M}$ where $x$ is the outcome of a trajectory starting at $(f, X)$ under policy $\pi$.*

**Impact on model architecture.** Although both the direct and BQ-MDPs are equivalent in terms of solving their associated COP, their practical interpretation leads to major differences. In the direct MDP view, it would not make sense to learn each instance-specific MDP separately. Instead a generic MDP conditioned on an input instance is learnt, similar to goal-conditioned Reinforcement Learning [40, 31] (the goal is here the input instance). A typical policy model architecture consists of an encoder in charge of computing an embedding of the input instance and a decoder that takes the instance embedding and the current partial solution to compute the next action (Fig. 1 left), e.g. the Attention Model [28] or PointerNetworks [45]. In the rollout of a trajectory, the encoder needs only be invoked once since the instance does not change throughout the rollout. For the BQ-MDP, only one, unconditional MDP is learnt for the whole solution space. The model can be simpler since the distinction between encoder and decoder vanishes (Fig. 1 right). On the other hand, the whole model must be applied to a new input instance at each step of a rollout.

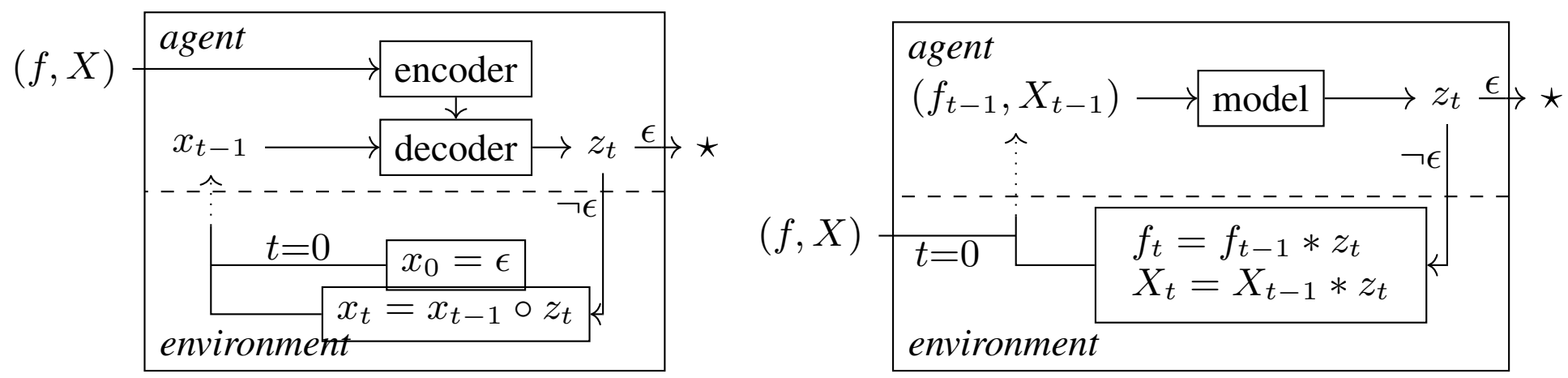

Figure 1: Policy model architectures for the direct MDP (left) and BQ-MDP (right).

### 3.3 Instance parametrization and recursion

For a given COP, instances are generally described by a set of parameters, which are used as input to the policy network. In the BQ-MDP, the instance is updated at each step $z$, according to the equations $X'{=}X{*}z$ and $f'{=}f{*}z$. In order to implement the BQ-MDP, a key requirement is that $(f', X')$ can be represented in the *same* parametric space as $(f, X)$. In fact, this is the case for COPs that satisfy the *tail-recursion property*: after applying a number of construction steps to an instance, the remaining tail subproblem is itself an instance of the original COP. This is a very common property in CO and includes in particular the Optimality Principle of Dynamic Programming [3, 6]: all problems that are amenable to dynamic programming satisfy the tail-recursion property. For these tail-recursive COPs, the bisimulation simply maps a partial solution to the tail subproblem instance it induces.

**Application to the KP and path-TSP.** One can see that the KP naturally satisfies the tail-recursion property. Consider an instance of the KP with capacity $c$ and items $(w_i, v_i)_{i\in I}$ described by their weight $w_i$ and value $v_i$. A partial solution is a subset $J{\subset}I$ of items. In this case, the bisimulation $\mathbf{\Phi}$ maps $J$ to a new KP instance with capacity $c{-}\sum_{j\in J} w_j$ and item set $(w_i, v_i)_{i\in I\setminus J}$ (see Appendix C). For the TSP, we introduce a slightly more general problem, path-TSP, where the goal is to find a shortest path between an origin and a destination node visiting a set of customer nodes, instead of a tour. The tail subproblem for a partial path $x_{1:k}$ consists of finding the shortest path from $x_k$ to the destination that visits all the remaining nodes, hence it is also a path-TSP instance. Thus path-TSP is tail-recursive, and TSP is simply a sub-problem of path-TSP where origin and destination are associated with the same Euclidean point (the depot). A similar reasoning holds for CVRP and OP, leading to path-CVRP and path-OP (see in Appendix A and B).

## 4 Policy Learning

We now describe our proposed transformer-based policy network for the BQ-MDPs. For simplicity, we focus here on the path-TSP; the models for path-CVRP, path-OP and KP differ only slightly and are presented in Appendix A, B and C.

**Neural architecture.** For the Euclidean path-TSP (Fig. 2 left), the Cartesian coordinates of each node (including origin and destination) are embedded via a linear layer. The remainder of the model is based on Vaswani et al. [43] with the following differences. First, we do not use positional encoding since the input nodes have no order. Instead, a learnable origin (resp. destination) encoding vector is added to the feature embedding of the origin (resp. destination) node. Second, we use ReZero [2]

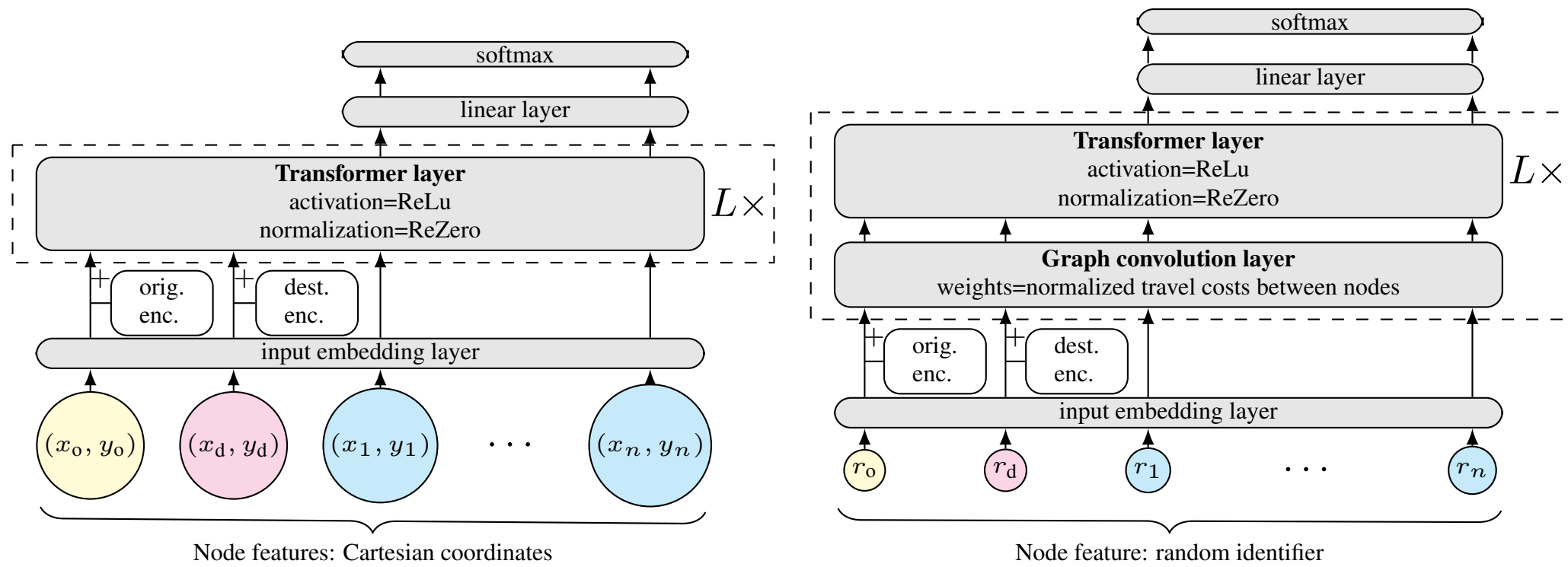

Figure 2: BQ-MDP policy model architecture for the TSP (left) and ATSP (right); origin in yellow, destination in red and the other nodes in blue.

normalization, which leads to more stable training and better performance in our experiments. Finally, a linear layer projects the output of the last attention block into a vector of size $N$, from which unfeasible actions, corresponding to the origin and destination nodes, are masked out, before applying a softmax operator so as to interpret the scalar node values for all allowed nodes as action probabilities. Note the absence of autoregressive decoder layers. In the asymmetric TSP, node coordinates are not given, but the travelling cost between nodes is provided through a (possibly asymmetric) matrix. The policy model for path-ATSP (Fig. 2 right) differs slightly from that for path-TSP. Initially, a randomly generated identifier is assigned to each node, which is embedded by a linear layer. The cost matrix is then incorporated by using a weighted graph convolution operation in each layer of the model before applying the attention mechanism. The edge weights of the graph are obtained by some standard normalization of the cost matrix. The use of random node identifiers as model input is justified in ATSP by the absence of node specific features. In problems where such features are available, a random identifier can still be added as extra node feature, as that often improves performance.

**Trajectory generation.** We train our model by imitation of expert trajectories, using a cross-entropy loss. Such trajectories are extracted from pre-computed (near) optimal solutions for instances of a relatively small and fixed size. Even for hard problems, we leverage the fact that it is generally possible to efficiently solve small instances (e.g. using MILP solvers). Of course the quality of the learned policy will be assessed on larger and therefore more challenging instances. Note that optimal solutions are not directly in the form of trajectories, i.e. sequences of construction steps. While Prop. 2 guarantees that a trajectory exists for any solution, it is usually not unique. In the TSP, an optimal tour corresponds to two possible trajectories (one being the reverse of the other). In the CVRP, each subtour similarly corresponds to two possible trajectories, and in addition, the different orders of the subtours lead to different trajectories. We noticed that this final order has an impact on the performance (see Appendix E.3). On the other hand, any sub-sequence from a fixed trajectory solution can be interpreted as the solution of a sub-instance. Note that these sub-instances will vary both in size and node distribution, therefore by training on them, we implicitly encourage the model to work well across sizes and node distributions, and generalize better than if such variations were not seen during the training.

**Complexity.** Because of the quadratic complexity of self-attention, and the fact that we call our policy at each construction step, the total complexity of our model, for an instance of size $N$, is $\mathcal{O}(N^3)$, whereas closely related transformer-based models such as the TransformerTSP [7] and the Attention Model [28] have a total complexity of $\mathcal{O}(N^2)$. We experimentally show in Sec. 6 that our model still provides better-quality solutions faster than these baselines. This is because most neural models are called many times per instance in practice, typically for sampling from the learned policy or within a beam search. In contrast, we observe that doing a single rollout of our policy yields excellent results. Intuitively, given the complexity (NP-hardness) of the targeted COPs, it makes sense to spend as much computation as possible at each time step, provided the whole rollout is fast enough. On the other hand, the quadratic attention bottleneck can be addressed by using some linear attention models. In our experiments, we show that replacing the Transformer by the linear PerceiverIO [19] significantly accelerates inference while still providing a competitive performance.

## 5 Related Work

**Generic frameworks for CO.** Generic frameworks have been crucial in the development and adoption of CO since they allow to formulate a COP in a certain format and then give it as an input to a (black box) solver. Among the most widespread frameworks are Mixed-Integer Programming [11] and Constraint Programming [39]. On the other hand, Markov Decision Processes are a powerful and ubiquitous tool to model sequential decision making problems and make them amenable to Machine Learning. While using MDPs to solve CO is not new, there are very few attempts to make it a general framework. Khalil et al. [23] makes a first attempt to define a generic MDP for greedy heuristics on graph problems. Drori et al. [13] also focuses on graph problems and provides a flexible and efficient GNN-based architecture to train with Reinforcement Learning (RL). Unlike these methods, our approach does not assume a graph or any structure on the COPs to define the (direct and BQ) MDP and we are the first to prove the equivalence between our proposed MDPs and solving the COP.

**Neural-based constructive heuristics.** Many NCO approaches construct solutions sequentially, via auto-regressive models. Starting with the seminal work by Vinyals et al. [45], which proposed the Pointer network trained in a supervised way, Bello et al. [4] trained the same model by RL for the TSP and Nazari et al. [34] adapted it for the CVRP. Kool et al. [28] introduced an Attention-based encoder-decoder Model (AM) trained with RL to solve several routing problems. This architecture was reused, with a few extensions, in POMO [29] and further combined with efficient search procedures in [10]. TransformerTSP [7] uses a similar architecture with a different decoder for the TSP. While most of these auto-regressive constructive approaches used RL, another line of work focuses on learning, in a supervised way, a heatmap of solution segments: Nowak et al. [35] trained a Graph Neural Network to output an adjacency matrix, which is converted into a feasible solution using beam search; Joshi et al. [21] followed a similar framework and trained a Graph Convolutional Network instead, that was used by [14] as well. Recently Sun and Yang [42] proposed DIFUSCO, a graph-based diffusion model to output the heatmap. Although such non-autoregressive methods are promising for large instances since they avoid the sequential decoding, they generally rely on sophisticated search methods (such as Monte Carlo Tree Search) in order to get a high-quality solution from the heatmap. Our paper explores the less common combination of supervised (imitation) learning to train a policy (not a heatmap) and achieves excellent performance without any search. In the context of planning, a similar strategy was successfully used in [16] to learn generalized reactive policies on small planning problems instances while generalizing well to larger ones.

**Step-wise approaches.** Instead of encoding an instance once and decoding sequentially as in the AM [28], Peng et al. [36] proposed to update the encoding after each subtour completion for CVRP while Xin et al. [47] updated it after each node selection in TSP. Xin et al. [46] extended on this idea by introducing the Multi-Decoder Attention Model (MDAM) which contains a special layer to efficiently approximate the re-embedding process. Since MDAM constitutes the most advanced version, we employ it as a baseline in our experiments.

**Generalizable NCO.** Generalization to different instance distributions, and esp. larger instances, is regarded as one of the major challenges for current NCO approaches [22, 33]. Fu et al. [14] trained a Graph Convolution model in a supervised manner on small graphs and used it to solve large TSP instances, by applying the model on sampled subgraphs and using an expensive MCTS search to improve the final solution (Att-GCN+MCTS). While this method achieves excellent generalization on TSP instances, MCTS requires a lot of computing resources and is essentially a post-learning search strategy. Geisler et al. [15] investigate the robustness of NCO solvers through adversarial attacks and find that existing neural solvers are highly non-robust to out-of-distribution examples and conclude that one way to address this issue is through adversarial training. In particular, Xin et al. [48] trains a GAN to generate instances that are difficult to solve for the current model. Manchanda et al. [32], Son et al. [41], Zhou et al. [50] take a different approach and leverage meta-learning to learn a model that is easily adaptable to new distributions. In DIMES, Qiu et al. [38] combines meta-learning with a continuous parametrization of the candidate solutions that allows to solve large-scale TSP instances. Accounting for symmetries in a given COP is a powerful idea to boost the generalization performance of neural solvers. For example, for problems on Euclidean graphs, Sym-NCO [24] make use of solution symmetries, to enforce robust representations while [29] exploits it as part of their loss function during training and at inference time to augment the set of solutions. Note that the above adversarial or meta-learning strategies and data augmentation are orthogonal to our approach of providing an efficient MDP which has beneficial effects irrespective of the training method. Finally,

some hybrid methods have been successful in solving large-scale routing problems [30, 51, 18]; these ate based on the divide-and-conquer principle: they learn how to split large instances into smaller ones, that are then efficiently solved by external specialized solvers.

## 6 Experimental Evaluation

To evaluate the effectiveness of our method, we test it on four routing problems - Euclidean TSP, CVRP, OP and Asymmetric TSP and one non-routing problem - KP (details and results for the latter are in Appendix C). For all routing problems, we train our model and all baselines on synthetic instances of size 100, following the same procedure of data generation as in [28]. For asymmetric TSP, we use Euclidean distance matrices and randomly break symmetries. We choose graphs of size 100 because it is the largest size for which (near) optimal solutions are still reasonably fast to obtain, and such training datasets are commonly used in the literature. Then, we test the trained models on synthetic instances of size 100, 200, 500 and 1K generated from the same distribution, as well as the standard TSPLib and CVRPLib datasets. Except for perceiver-based models and OP (dist), we limit the subgraphs to the 250 nearest neighbours of the origin node at test time. This reduces inference time while not significantly impacting the solution quality (see Appendix D).

**Hyperparameters and training.** We use the same hyperparameters for all problems. The model has 9 layers, each built with 12 attention heads with embedding size of 192 and dimension of feed-forward layer of 512. Our model is trained by imitation of expert trajectories, using a cross-entropy loss. Solutions of these problems are obtained by using the Concorde solver [1] for TSP, LKH heuristic [17] for ATSP and CVRP, and EA4OP heuristic [26] for OP. We use a dataset of 1 million solutions. To sample trajectories out of this dataset, we note that in the case of TSP, any sub-path of the optimal tour is also an optimal solution to the associated path-TSP sub-problem, hence amenable to our path-TSP model. We therefore form minibatches by first sampling a number $n$ between $4$ and $N$ (path-TSP problems with less than $4$ nodes are trivial), then sampling sub-paths of length $n$ from the initial solution set. At each epoch, we sample a sub-path from each solution. By sampling subsequences among all possible infixes of the optimal solutions, we automatically get an augmented dataset, which some previous models had to explicitly design their model for [29]. We use a similar sampling strategy for CVRP, OP and KP (see Appendix .A, B and C). Batches of size 1024 are formed, and the model is trained for 500 epochs. We use Adam [25] as optimizer with an initial learning rate of $7.5\text{e}{-4}$ and decay of 0.98 every 50 epochs.

**Baselines and test datasets.** We compare our model with the following methods: OR-Tools [37], LKH [17], and Hybrid Genetic Search (HGS) for the CVRP [44] as SOTA non-neural methods; DIFUSCO+2opt, Att-GCN+MCTS and SGBS as hybrid methods; and AM, TransformerTSP, MDAM, POMO, DIMES and Sym-NCO as deep learning-based constructive methods. Description of all baseline models are provided in Sec. 5. For all deep learning baselines we use the corresponding model *trained on graphs of size 100* and the best decoding strategy[2]. Following the same procedure as in [14], we generate four test datasets with graphs of sizes 100, 200, 500 and 1000. For CVRP, we use capacities of 50, 80, 100 and 250, respectively. In addition, we report the results on TSPLib instances with up to 4461 nodes and all CVRPLib instances with node coordinates in the Euclidean space. For all models, we report the optimality gap and the inference time. The optimality gap for TSP is based on the optimal solutions obtained with Concorde. For CVRP, although HGS gives better results than LKH, we use the LKH solutions as a reference to compute the "optimality" gap, in order to be consistent (and easily comparable) with previous works. For the OP task, we train and test the model using a "distance" variant of the problem, as described in [28]. For fixed maximum lengths, we chose $T^{100} = 4$, $T^{200} = 5$, $T^{500} = 6$, and $T^{1000} = 7$. While the optimality gap is easy to compute and compare, running times are harder to compare since they strongly depend on the implementation platforms (Python, C++), hardware (GPU, CPU), parallelization, batch size, etc. In our experiments, we run all deep learning models on a single Nvidia Tesla V100-S GPU with 24GB memory, and other solvers on Intel(R) Xeon(R) CPU E5-2670 with 256GB memory, in one thread.

**Results.** Table 1 summarizes our results for Euclidean TSP, CVRP and OP as well as asymmetric TSP. For all problems, our model shows superior generalization on larger graphs, even with a

[2]Att-GCN+MCTS is originally trained on graph sizes of 50. For DIMES and DIFUSCO authors provide generalization tables just for two types of decoding (greedy+2opt for DIFUSCO and RL+sampling for DIMES) and on some test datsets, so we report them in our results

greedy decoding strategy, which generates a single solution while all other baselines generate several hundreds (and select the best among them). In terms of running time, with greedy decoding, our model is slightly slower that Sym-NCO baseline, competitive with POMO, and significantly faster than other models. Beam search further improves the optimality gaps, but as expected, it takes more time. On the other hand, our model can be accelerated by replacing the quadratic attention blocks by the PerceiverIO architecture [19] (more details in Appendix E.1). This results in a considerable reduction of inference time for larger graphs, at the cost of some performance drop. Even with this trade-off, our model achieves remarkable performance compared with other NCO baselines. Figure 3 shows optimality gap versus running time for our models and other baseline models. Our models clearly outperform other models in terms of generalization to larger instances. The only models that are competitive with ours are Att-GCN+MCTS and DIFUSCO+2opt, but both are hybrid methods with an expensive search on top of the output of the neural network, and they are 2 to 15 times slower while being designed for TSP only. Finally, in addition to the synthetic datasets, we test our models on TSPLib and VRPLib instances, which are of varying graph sizes, node distributions, demand distributions and vehicle capacities. Table 1 shows that our model significantly outperforms the end-to-end baseline policies even with the greedy decoding strategy.

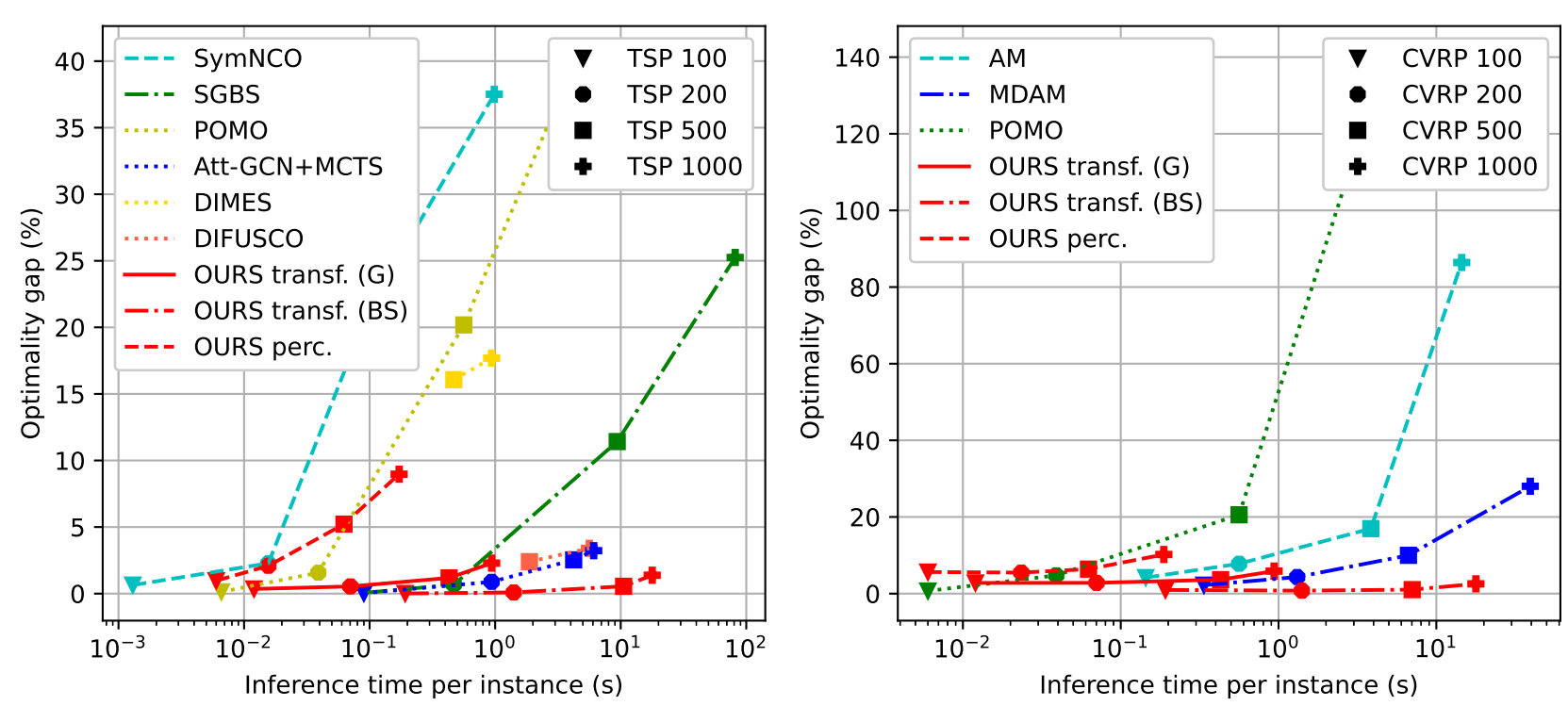

Figure 3: Generalization results on different graph sizes for TSP (left) and CVRP (right). Lower and further left is better.

## 7 Conclusion

We proposed a principled framework to formulate COPs as efficient symmetry-invariant MDPs. First, we presented how to derive a direct MDP for any COP that is amenable to a constructive approach, and proved the exact correspondence between the optimal policies of this direct MDP and the optimal solutions of the COP it stems from. Second, for the widespread class of recursive COPs, we introduced bisimulation quotienting to leverage the symmetries of the partial solutions. We showed that the resulting BQ-MDP has a significantly reduced state space while its optimal policies still exactly solve the original COP. We then designed a simple Transformer-based architecture as an examplification of a possible policy network for the BQ-MDPs, and applied it to five classical COP problems: (A)TSP, CVRP, OP and KP. The resulting policy generalizes particularly well, and significantly outperforms not only similar end-to-end neural-based constructive solvers but also hybrid methods that combine learning with search. A limitation of our approach, shared with all constructive methods, is that it can only handle problems for which feasibility is easy to ensure throughout the construction process. In order to address problems with more challenging global constraints, approaches which combine learning and constraint programming (e.g. [8]) may be a promising direction. While training on relatively small instances allowed us to use imitation learning, our proposed MDPs and the associated model could also be trained with Reinforcement Learning. This would be particularly interesting to extend our framework from deterministic to stochastic COPs.

| | | Test (10k inst.) | | Generalization (128 instances) | | | | | |
|---|---|---|---|---|---|---|---|---|---|
| | | N=100 | | N=200 | | N=500 | | N=1000 | |
| | | opt. gap | time | opt. gap | time | opt. gap | time | opt. gap | time |
| TSP | Concorde | 0.00% | 38m | 0.00% | 2m | 0.00% | 40m | 0.00% | 2.5h |
| | OR-Tools | 3.76% | 1.1h | 4.52% | 4m | 4.89% | 31m | 5.02% | 2.4h |
| | Att-GCN+MCTS* | 0.04% | 15m | 0.88% | 2m | 2.54% | 6m | 3.22% | 13m |
| | DIFUSCO G+2opt* | 0.24% | - | - | - | 2.40% | 4m | 3.40% | 12m |
| | SGBS (10,10) | 0.06% | 15m | 0.67% | 1m | 11.42% | 20m | 25.25% | 2.9h |
| | AM bs1024 | 2.51% | 20m | 6.18% | 1m | 17.98% | 8m | 29.75% | 31m |
| | TransTSP bs1024 | 0.46% | 51m | 5.12% | 1m | 36.14% | 9m | 76.21% | 37m |
| | MDAM bs50 | 0.39% | 45m | 2.04% | 3m | 9.88% | 13m | 19.96% | 1.1h |
| | POMO augx8 | 0.13% | 1m | 1.57% | 5s | 20.18% | 1m | 40.60% | 10m |
| | DIMES RL+S* | - | - | - | - | 16.07% | 1m | 17.69% | 2m |
| | Sym-NCO s100 | 0.64% | 13s | 2.28% | 2s | 21.64% | 13s | 37.51% | 2m |
| | **BQ-perceiver G** | 0.97% | 1m | 2.09% | 2s | 5.22% | 8s | 8.97% | 22s |
| | **BQ-transformer G** | 0.35% | 2m | 0.54% | 9s | 1.18% | 55s | 2.29% | 2m |
| | **BQ-transformer bs16** | **0.01**% | 32m | **0.09**% | 3m | **0.55**% | 15m | **1.38**% | 38m |
| CVRP | LKH | 0.00% | 15.3h | 0.00% | 30m | 0.00% | 1.3h | 0.00% | 2.8h |
| | HGS | -0.51% | 15.3h | -1.02% | 30m | -1.25% | 1.3h | -1.10% | 2.8h |
| | OR-Tools | 9.62% | 1h | 10.70% | 3m | 11.40% | 18m | 13.56% | 43m |
| | AM bs1024 | 4.18% | 24m | 7.79% | 1m | 16.96% | 8m | 86.41% | 31m |
| | MDAM bs50 | 2.21% | 56m | 4.33% | 3m | 9.99% | 14m | 28.01% | 1.4h |
| | POMO augx8 | 0.69% | 1m | 4.77% | 5s | 20.57% | 1m | 141.06% | 10m |
| | Sym-NCO s100 | 1.46% | 15s | 4.84% | 3s | 18.64% | 13s | 119.60% | 9m |
| | SGBS (4,4) | **0.08%** | 15m | 2.47% | 1m | 14.98% | 20m | 151.07% | 2.9h |
| | **BQ-perceiver G** | 5.63% | 1m | 5.49% | 3s | 6.39% | 8s | 10.21% | 24s |
| | **BQ-transformer G** | 2.79% | 2m | 2.81% | 9s | 3.64% | 55s | 5.88% | 2m |
| | **BQ-transformer bs16** | 0.95% | 32m | **0.77**% | 3m | **1.04%** | 15m | **2.55%** | 38m |
| OP (DIST.) | EA4OP | 0.00% | 1.4h | 0.00% | 3m | 0.00% | 11m | 0.00% | 45m |
| | AM bs1024 | 2.19% | 17m | 8.04% | 1m | 20.73% | 4m | 29.96% | 15m |
| | MDAM bs50 | 0.69% | 32m | 2.98% | 49s | 15.36% | 4m | 26.55% | 15m |
| | SymNCO s200 | 0.45% | 4m | 3.39% | 2s | 15.23% | 37s | 26.95% | 2m |
| | **BQ-transformer G** | -0.15% | 2m | 0.62% | 8s | 3.26% | 42s | 8.40% | 8.9m |
| | **BQ-transformer bs16** | **-1.19**% | 35m | **-0.84**% | 2.1m | **1.04%** | 21m | **4.81%** | 2.2h |
| ATSP | LKH | 0.00% | 18m | 0.00% | 29s | 0.00% | 2.3m | 0.00% | 9.4m |
| | MatNet s128† | **0.93**% | 45m | 124.20% | 4m | - | - | - | |
| | **BQ-GCN-transformer G** | 1.27% | 1m | 1.56% | 4s | 4.22% | 12s | 11.59% | 14s |
| | **BQ-GCN-transformer bs16** | 0.96% | 19m | **1.41**% | 1m | **2.43**% | 3m | **8.26**% | 7m |

Table 1: BQ-models with greedy rollouts (G) or Beam Search (bs) versus classic and neural baselines. The results for models* are taken from the original papers. For a fair comparison, model † are evaluated on instances generated from the same distribution as the training datasets for this model, which may differ from our training and/or test distribution.

| | MDAM | POMO | BQ (ours) | |
|---|---|---|---|---|
| Size | bs50 | x8 | greedy | bs16 |
| <100 | 3.06% | 0.42% | 0.34% | **0.06**% |
| 100-200 | 5.14% | 2.31% | 1.99% | **1.21**% |
| 200-500 | 11.32% | 13.32% | 2.23% | **0.92**% |
| 500-1K | 20.40% | 31.58% | 2.61% | **1.91**% |
| >1K | 40.81% | 62.61% | 6.42% | **5.90**% |
| All | 19.01% | 26.30% | 3.30% | **2.55%** |

| | MDAM | POMO | BQ (ours) | |
|---|---|---|---|---|
| Set (size) | bs50 | augx8 | greedy | bs16 |
| A (32-80) | 6.17% | 4.86% | 4.52% | **1.18**% |
| B (30-77) | 8.77% | 5.13% | 6.36% | **2.48**% |
| F (44-134) | 16.96% | 15.49% | 6.63% | **5.94**% |
| M (100-200) | 5.92% | 4.99% | 3.58% | **2.66**% |
| P (15-100) | 8.44% | 14.69% | 3.61% | **1.54**% |
| X (100-1K) | 34.17% | 21.62% | 9.94% | **7.22**% |
| All (15-1K) | 22.36% | 15.58% | 7.62% | **4.83**% |

Table 2: Experimental results on TSPLib (left) and CVRPLib (right).

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
