# A  Application to the Capacitated Vehicle Routing Problem

**Problem definition.**    The Capacitated Vehicle Routing Problem (CVRP) is a vehicle routing problem in which a vehicle with limited *capacity* must deliver items from a depot location to various customer locations. Each customer has an associated *demand*, and the goal is to compute a set of subtours for the vehicle, starting and ending at the depot, such that all the customers are visited, the sum of the demands per subtour of the vehicle does not exceed the capacity, and the total travelled distance is minimized.

**Solution space.**    Formally, a partial solution (in $\mathcal{X}$) for CVRP, just as for TSP, is a finite sequence of nodes. Similarly, the ○ operator is sequence concatenation and the neutral element $\epsilon$ is the empty sequence. In a CVRP instance, as in TSP, each node is assigned a location, and the objective function $f$ for an arbitrary sequence of nodes is the total travelled distance for a vehicle visiting the nodes in sequence:

$$f(x_{1:n}) \;=\; \sum_{i=2}^{n} \mathrm{dist}(x_{i-1}, x_i).$$

The feasible set $X$ consists of the sequences $x_{1:n}$ of nodes which start and end at the depot, which are pairwise distinct except for the depot, and such that the cumulated demand of any contiguous subsequence $x_{i:j}$ not visiting the depot, i.e. a segment of a subtour, does not exceed the capacity of the vehicle:

$$x_{1:n} \in X \text{ iff } \begin{cases} x_1 = x_n = \text{depot}, \\ \forall i,j \in \{1{:}n\} \quad x_i = x_j \neq \text{depot} \implies i = j, \\ \forall i,j \in \{1{:}n\} \quad \forall k \in \{i{:}j\}\; x_k \neq \text{depot} \implies \sum_{k=i}^{j} \text{demand}(x_k) \leq \text{capacity}. \end{cases}$$

Just as with TSP, CVRP on its own does not satisfy the tail-recursion property of Sec. 3.3, but is a particular case of a more general problem called *path*-CVRP (similar to path-TSP) which does satisfy that property. In path-CVRP, instead of starting at the depot with its full capacity, the vehicle starts at an *origin* node with a given *initial capacity*. A CVRP instance is a path-CVRP instance where origin and depot are the same and the initial capacity is the full capacity. In path-CVRP, each tail subproblem after selection of a node $z$ updates both the origin (which becomes $z$) and the initial capacity (which is decremented by the demand at $z$ if $z$ is a customer node or reset to the full capacity if $z$ is the depot), conditioned on the resulting capacity being non negative.

**Bisimulation quotienting.**    Given the above solution space, we can directly apply the definitions of Sec 3.2 to define the bisimulation and the resulting BQ-MDP for path-CVRP, illustrated below:

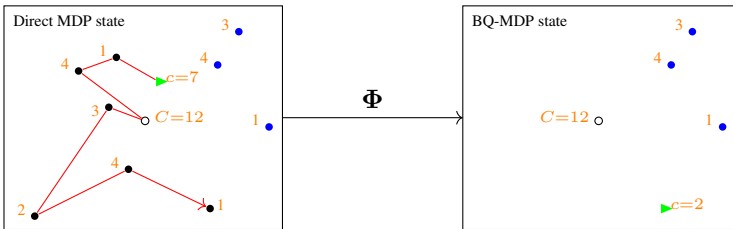

The diagram on the left represents a direct MDP state, i.e. a path-CVRP instance together with a partial solution. The depot is represented as a white disk with its full capacity ($C{=}12$), the origin as green triangle with initial capacity ($c{=}7$), customer nodes as blue or black disks with their demands. The partial solution is the sequence of black nodes represented by the (directed) red path. The diagram on the right represents the corresponding BQ-MDP state, i.e. a path-CVRP instance. Note the new origin (the end node of the partial solution), and its initial capacity $c{=}2$: it is the full capacity $C{=}12$ minus the cumulated demand served since the last visit to the depot ($3{+}2{+}4{+}1$)

**Model architecture.**    The model architecture for the CVRP is almost the same as for the TSP, with a slight difference in the input and output layers. In the TSP model, the input to the node embedding layer for a $N$-node state is a $2{\times}N$ matrix of coordinates. For CVRP, we use two additional channels: one for the node's demand, and one for the current vehicle capacity, repeated across all nodes. The demand is set to zero for the origin and depot nodes. We obtain a $4{\times}N$ matrix of features, which is

passed through a learned embedding layer. As for the TSP, a learned origin (resp. depot) encoding vector is added to the corresponding node embeddings. The rest of the architecture, in the form of attention layers, is identical to TSP, until after the action scores projection layer. In the case of TSP, the projection layer returns a vector of $N$ scores, where each score, after a softmax, represents the probability of choosing that node as the next step in the construction. In the case of CVRP, the model returns a matrix of scores of dimension $N \times 2$, corresponding to each possible actions and the softmax scopes over this whole matrix. An action is here either the choice of the next node, as in TSP, or of the next two nodes, the first one being the depot As usual, a mask is always applied to unfeasible actions before the softmax operator: those which have higher demand than the remaining vehicle capacity, as well as the origin and depot nodes.

## B    Application to the Orienteering Problem

**Problem definition.**    The Orienteering Problem (OP) is a combinatorial optimization problem in which we need to find the optimal route to visit a set of given locations within a given distance (or time) limit. The route must start and end at a given location (usually called a depot), each location is associated with a scalar prize, and the goal is to maximize the cumulated prize, respecting the distance (time) constraint. This problem has applications in various fields, including logistics and planning.

**Solution space and bisimulation quotienting.**    After defining BQ-MDP for path-TSP and path-CVRP, defining BQ-MDP for path-OP is straightforward. The partial solution is the sequence of already visited nodes, together with remaining distance limit. An OP instance is a path-OP instance where origin and destination are the same (depot) and distance limit equals the initial distance constraint. Just as in path-TSP and path-CVRP, in path-OP, each tail subproblem, after selecting a node $z$, updates both the origin (which becomes $z$) and the remaining distance constraint (the current distance constraint is decreased by the distance from the previous origin to the selected node $z$). However, in OP, unlike TSP and CVRP, the number of steps is not known in advance - for TSP and CVRP, construction of solution ends when all locations are visited, whereas in OP, it ends when a given distance budget is exceeded.

**Model architecture.**    The model architecture for path-OP is the same as for path-TSP, with two additional input channels for node prize and distance constraint (repeated across all nodes, as the total capacity in CVRP). Thus, we obtain a $4 \times N$ matrix of features (two for coordinates, one for prize and one for distance constraint), which is passed through a learned embedding layer. As usual, a learned encoding for origin and depot nodes are added to the corresponding node embeddings. The rest of the architecture, including output projection layer is the same as for TSP. Before applying the softmax operator, we apply a mask to exclude origin and depot nodes, as well as nodes that cannot be visited due to the distance constraint.

## C    Application to the Knapsack Problem

The knapsack problem and its solution space were described in Sec. 2.1; the associated bisimulation in Sec 3.3. We provide below an illustration of the bisimulation on an example:

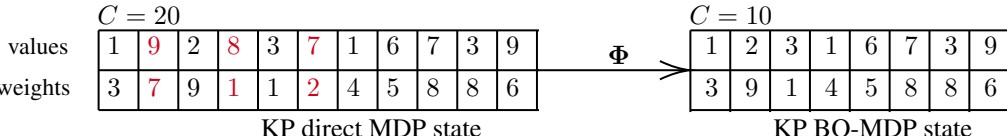

Here, the knapsack capacity is $C = 20$ and each item is defined by its weight (bottom cell) and value (top cell). Mapping $\Phi$ for KP is straightforward: it removes all picked items and updates the remaining capacity by subtracting total weight of removed items from the previous capacity.

**Model architecture.**    The model architecture for KP is again very similar to previously described models for TSP, CVRP and OP. The input to the model is a $3 \times N$ tensor composed of items features (values, weights) and the additional channel for the remaining knapsack's capacity. By definition, the solution has no order (the result is a set of items), so there is no need to add tokens for origin and

|  |  | Optimal | POMO (single traj.) | | POMO (all traj.) | | BQ **ours** (greedy) | |
|---|---|---|---|---|---|---|---|---|
|  |  | value | value | optgap | value | optgap | value | optgap |
| N=200 | C=10 | 36.073 | 34.062 | 5.565% | 34.961 | 3.076% | **35.961** | **0.311**% |
|  | C=25 | 57.429 | 57.143 | 0.499% | **57.420** | **0.016**% | 57.371 | 0.102% |
|  | C=50 | 81.100 | 79.766 | 1.617% | 80.085 | 1.229% | **80.564** | **0.668**% |
|  | C=100 | 99.773 | 99.416 | 0.358% | 99.483 | 0.291% | **99.694** | **0.080**% |
| N=500 | C=10 | 57.456 | 51.829 | 9.769% | 54.213 | 5.627% | **56.853** | **1.054**% |
|  | C=25 | 91.026 | 85.186 | 6.414% | 86.482 | 4.992% | **90.741** | **0.314**% |
|  | C=50 | 128.999 | 128.646 | 0.273% | **128.946** | **0.042**% | 128.906 | 0.072% |
|  | C=100 | 182.395 | 181.615 | 0.424% | **181.870** | **0.285**% | 181.654 | 0.407% |
| N=1000 | C=10 | 81.334 | 53.319 | 34.401% | 58.072 | 28.565% | **79.650** | **2.074**% |
|  | C=25 | 128.993 | 122.112 | 5.340% | 123.775 | 4.046% | **128.240** | **0.584**% |
|  | C=50 | 182.813 | 170.223 | 6.877% | 171.789 | 6.021% | **181.985** | **0.451**% |
|  | C=100 | 257.411 | 252.701 | 1.831% | 253.361 | 1.575% | **257.224** | **0.072**% |
| All |  | - |  | 6.131% |  | 4.647% |  | **0.516**% |

Table 3: Average values and optimality gaps for KP on various instance distributions

destination. Apart from excluding these tokens and different input dimensions, the rest of the model is identical to the TSP model. The output is a vector of $N$ probabilities over all items with a mask over the unfeasible ones (with weights larger than remaining knapsack's capacity). In the training, at each construction step, any item of the ground-truth solution is a valid choice. Therefore we use a multi-class cross-entropy loss.

**Experimental results for KP.** We generate the training dataset as described in [29]. We train our model on 1M KP instances of size 200 and capacity 25, with values and weights randomly sampled from the unit interval. We use the dynamic programming algorithm from ORTools to compute the ground-truth optimal solutions. As hyperparameters, we use the same as for the TSP, except the training is shorter - it converges in just 50 epochs. Then, we evaluate our model on test datasets with 200, 500 and 1000 items and capacity of 10, 25, 50 and 100, for each problem size. Table 3 shows the performance of our model compared to POMO, one of the best performing NCO models on KP. Although our model does not outperform it on all datasets, it achieves better overall performance and significantly better performance on the out-of-distribution datasets (datasets of size 1000 and datasets with a capacity of 10). It should be noted again that POMO builds $N$ solutions per instance and chooses the best one, while our model generates a single solution per instance but still achieves better results.

## D   Impact of k-nearest-neighbor heuristic on model performance

As mentioned in the Sec. 6, inference time of our model can be reduced by using a $k$-nearest-neighbor heuristic to restrict the search space. For both greedy rollouts and beam search strategies, at every step, it is possible to reduce the remaining graph by considering only a certain number of neighboring nodes. Table 4 presents the experiments on TSP and CVRP where we apply the model on different number of nearest neighbors of the origin. This approach clearly reduces the execution time, but also in some cases even improves the performance in terms of optimality gap. Note that the criteria on which to select the nearest neighbors does not have to be the distance but the same metric as some greedy heuristic for the problem. For example for the Knapsack problem, the items could be restricted to the $k$ items with highest values (or highest ratios of value/weight).

## E   Ablation study

### E.1   PerceiverIO architecture

To construct a solution, our model needs to perform $N$ steps, and compute $N^2$ attention matrices at each step, so total complexity is $\mathcal{O}(N^3)$. Although our model outperforms current state-of-the-art models in terms of both performance and inference time, this may become a limiting factor when applying the model to large graph sizes. This is a well-known issue for all attention models, and

| | | TSP | | | | CVRP | | | |
|---|---|---|---|---|---|---|---|---|---|
| | | Greedy | | Beam size 16 | | Greedy | | Beam size 16 | |
| N=500 | full graph | 1.091% | 2m | 0.572% | 28m | 3.951% | 2m | 1.503% | 28m |
| | 250 KNNs | 1.186% | 1m | 0.550% | 15m | 3.645% | 1m | 1.040% | 15m |
| N=1000 | full graph | 2.141% | 14m | 1.412% | 3.5h | 6.282% | 15m | 3.660% | 5.4h |
| | 500 KNNs | 2.086% | 7m | 1.348% | 2.7h | 6.330% | 10m | 3.382% | 2.7h |
| | 250 KNNs | 2.294% | 2m | 1.379% | 38m | 5.883% | 2m | 2.552% | 38m |

Table 4: Experimental results with different numbers $N$ of nearest-neighbors during the inference.

there have been various proposals to reduce the complexity of attention. In this work, we propose a compromise between model complexity and quality of the solution by replacing the standard transformer model with the PerceiverIO architecture [19]. PerceiverIO computes cross-attention between input data and latent variables and then compute self-attention between the latent variables, resulting in all computations being done in the latent space. This approach allows the number of operations to be linear (instead of quadratic) in the input's length.

In our implementation, we use similar hyperparameters as for the transformer model: 9 attention layers with 8 heads, an embedding size of 192, and a feed-forward layer dimension of 512. For the latent space, we use a vector with dimensions of $64 \times 48$, while the output query array is the same as the input array.

## E.2    Approximated model

As mentioned in Section 5, existing works have also noted the importance of accounting for the change of the state after each action: [47, 46] claimed that models should recompute the embeddings after each action. However because of the additional training cost, they proposed the following approximation: fixing lower encoder levels and recomputing just the top level with a mask of already visited nodes. They hypothesis a kind of hierarchical feature extraction property that may make the last layers more important for the fine-grained next decision. In contrast, we call our entire model after each construction step; effectively recomputing the embeddings of each state. We hypothesis that this property may explain the superior performance (Table 1) w.r.t MDAM model [46]. In order to support this hypothesis, we have implemented an approximated version of our model as follows. We fixed the bottom layers of our model and recomputed just the top layer, by masking already visited nodes and adding the updated information (origin and destination tokens for TSP). As expected, inference time is 1.6 times shorter, but performance is severely degraded: we obtained optimality gap of 8.175% (vs 0.35% with original model) on TSP100.

## E.3    On the impact of expert trajectories

Our datasets consist of pairs of a problem instance and a solution. For imitation learning, we need pairs of a problem instance and an expert trajectory in the MDP. However multiple trajectories may be obtained from the solution. For example, in the TSP, a solution is a loop in a graph, and one has to decide at which node its construction started and in which direction it proceeded. In the CVRP, the order in which the subtours are constructed needs also to be decided. Hence, all our datasets are pre-processed to transform solutions into corresponding construction trajectories (a choice for each or even all possible ones). Our experiments demonstrate that this transformation has a significant impact on performance. Specifically, in the CVRP, we found that the best performance is achieved by training the model on expert solutions that sort subtours by the remaining vehicle capacity at the end of each subtour. More precisely, the last subtour in the expert trajectory has the biggest remaining capacity (the subtour that visits remaining unvisited nodes), while the first subtour had the smallest remaining capacity (usually 0). This simple data preprocessing step leads to an almost twofold improvement in performance compared to training on expert trajectories with subtours in arbitrary order. Intuitively, these trajectories encourage the model to create subtours that use the whole vehicle capacity whenever possible.

# F Proofs

## F.1 Properties of the direct MDP

*Proof.* Let $\mathcal{X}$ be a solution space, $(f, X) \in \mathcal{F}_\mathcal{X}$ an instance and $\mathcal{M}_{(f,X)}$ its direct MDP. $\mathcal{X}$ is normally designed to accommodate all the instances of a CO problem, possibly even of multiple problems, so we do not make the assumption that $\mathcal{X}$ or $\mathcal{Z}$ are finite, but only that the conditions of Def. 1 are satisfied.

**The empty partial solution $\epsilon$ is a valid state**
By definition, $(f, X) \in \mathcal{F}_\mathcal{X}$ implies $X \neq \emptyset$, which is equivalent, by definition, to $\epsilon \in \bar{X}$, i.e. $\epsilon$ is in the state space of $\mathcal{M}_{(f,X)}$.

**Each state has a finite, non null number of allowed actions**.
Let $x \in \bar{X}$ be a state of $\mathcal{M}_{(f,X)}$.

By definition $x \circ y \in X$ for some $y \in \mathcal{X}$, and by (1), we have $y = z_1 \circ \cdots \circ z_n$ for some $z_{1:n} \in \mathcal{Z}$. If $n > 0$, then, by associativity, $(x \circ z_1) \circ z_2 \circ \cdots \circ z_n = x \circ y \in X$, hence by definition $x \circ z_1 \in \bar{X}$, hence step action $z_1$ is allowed from $x$. If $n = 0$ then $y = \epsilon$ and $x = x \circ y \in X$ hence the null action $\epsilon$ is allowed from $x$. In both cases, $x$ is not a dead end state.

Since $X$ is finite and, by (1), each of its elements has a finite number of step decompositions, the set $Z$ of steps occurring in at least one step decomposition of an element of $X$ is itself finite, even if $\mathcal{Z}$ is infinite. Now, assume $z$ is an allowed step action from $x$, hence, by definition $x \circ z \in \bar{X}$ hence $x \circ z \circ y \in X$ for some $y \in \mathcal{X}$. By (1), both $x$ and $y$ have step decompositions, hence $z$ occurs in at least one step decomposition of an element of $X$, i.e. $z \in Z$. Hence, all step actions allowed from a valid state are in the finite set $Z$.

**All but a finite number of transitions in a trajectory are null actions**
By definition and (1), an element of $\bar{X}$ is the composition of a prefix of a step decomposition of an element of $X$. Since $X$ is finite and, by (1), each of its elements has a finite number of step decompositions, $\bar{X}$ is finite.

Let $x_0 a_1 x_1 a_2 x_2 \cdots$ be an infinite trajectory of $\mathcal{M}_{(f,X)}$. By definition of the allowed transitions, it is easy to show that $x_n = x_0 \circ a_1 \circ \cdots \circ a_n$ for all $n \in \mathbb{N}$ and furthermore $x_n \in \bar{X}$. Let $N = \{n \in \mathbb{N} | a_n \neq \epsilon\}$. Reason by contradiction and assume $N$ is infinite, i.e. there exists an increasing sequence $n_{1:\infty}$ such that $N = \{n_i\}_{i=1}^\infty$. For all $i \geq 1$, we have $x_{n_i} = x_0 \circ z_1 \circ \cdots \circ z_i$ where $z_i = a_{n_i} \in \mathcal{Z}$ and $x_{n_i} \in \bar{X}$. Since $\bar{X}$ is finite, there exist indices $j > i$ such that $x_{n_j} = x_{n_i}$. Hence $x_{n_i} = x_{n_i} \circ z_{i+1} \circ \cdots \circ z_j$. Hence, any step decomposition of $x_{n_i}$ can be expanded with the non empty sequence $z_{i+1:j}$ and remain a step decomposition of $x_{n_i}$. Hence there are infinitely many step decompositions of $x_{n_i}$. Contradiction. Therefore, $N$ is finite. $\qquad\square$

## F.2 Soundness of the direct MDP

*Proof.* We first show the following general lemma. Let $Y \xrightarrow{\psi} X \xrightarrow{f} \mathbb{R} \cup \{\infty\}$ be arbitrary mappings. If $\psi$ is surjective then

$$\arg \min_{x \in X} f(x) = \psi(\arg \min_{y \in Y} f(\psi(y))) \qquad (4)$$

This is shown by simple application of the definition of $\arg \min$ (as a set). The subscript $_*$ denotes the steps where the assumption that $\psi$ is a surjection is used:

$$x' \in \psi(\arg \min_y f(\psi(y))) \quad \text{iff} \quad \exists y' \in \arg \min_y f(\psi(y)) \ x' = \psi(y')$$
$$\text{iff} \quad \exists y' \ x' = \psi(y') \ \forall y \ f(\psi(y')) \leq f(\psi(y)) \quad \text{iff} \quad \exists y' \ x' = \psi(y') \ \forall y \ f(x') \leq f(\psi(y))$$
$$\text{iff}_* \quad \forall y \ f(x') \leq f(\psi(y)) \quad \text{iff}_* \quad \forall x \ f(x') \leq f(x) \quad \text{iff} \quad x' \in \arg \min_x f(x)$$

Now, let $\mathcal{X}$ be a solution space, $(f, X) \in \mathcal{F}_\mathcal{X}$ an instance and $\mathcal{M}_{(f,X)}$ its direct MDP. Let $Y$ be the set of trajectories in $\mathcal{M}_{(f,X)}$ starting at $\epsilon$, and for each $y \in Y$, let $\psi(y)$ denote its outcome.

Observe that $\psi : Y \mapsto X$. Indeed, it has been shown above that a valid trajectory $y$ always ends with an infinite (stationary) sequence of null transitions on a state $x$ which is also its outcome when $y$ starts with $\epsilon$, i.e. $x = \psi(y)$. For the null transitions to be allowed, we must have $x \in X$.

Let's show that $\psi$ is surjective. Let $x \in X$. By (1), $x = z_1 \circ \cdots \circ z_n$ for some $z_{1:n} \in \mathcal{Z}$. For each $m \in \{0:n\}$, let $x_m = z_1 \circ \cdots \circ z_m$ and $x'_m = z_{m+1} \circ \cdots \circ z_n$. Hence $x_m \circ x'_m = x \in X$ hence $x_m \in \bar{X}$ is a

valid state. Hence, the sequence $y = x_0 z_1 x_1 \cdots z_n x_n (\epsilon x_n)^*$ is a valid trajectory of $\mathcal{M}_{(f,X)}$, and it is starting at $x_0 = \epsilon$, hence $y \in Y$. Its stationary state is $x_n = x$ hence $\psi(y) = x$. Hence $\psi$ is surjective.

Now, let $y = x_0 a_1 x_1 \cdots a_n x_n \cdots$ be a trajectory in $Y$. By definition, $x_0 = \epsilon$. Furthermore, by definition of the direct MDP transitions, the reward for $a_n$ (whether it is a step or null) is $f(x_{n-1}) - f(x_n)$, which is null when the trajectory becomes stationary on state $\psi(y)$. By summation, the total reward $R(y)$ of trajectory $y$ is $f(\epsilon) - f(\psi(y))$. The objective is defined up to an additive constant, so we can assume without loss of generality that $f(\epsilon) = 0$. Hence $f(\psi(y)) = -R(y)$.

Finally, since $\psi$ is surjective, we can apply (4) proved above and get

$$\arg\min_{x \in X} f(x) = \psi(\arg\min_{y \in Y} f(\psi(y))) = \psi(\arg\min_{y \in Y} -R(y)) = \psi(\arg\max_{y \in Y} R(y))$$

In other words, an optimal solution to $(f, X)$ is the outcome of an optimal trajectory of $\mathcal{M}_{(f,X)}$ starting at $\epsilon$, i.e. one which can be obtained by application of an optimal policy. $\qquad \square$

### F.3 Bisimulation between the direct MDP and the BQ-MDP

Let $\mathcal{X}$ be a solution space, $(f, X) \in \mathcal{F}_{\mathcal{X}}$ be an instance, $x \in \bar{X}$ be a valid state for the direct MDP of $(f, X)$. We have to show that $\Phi_{(f,X)}$ is a bisimulation $\mathcal{M}_{(f,X)} \leftrightarrow \mathcal{M}$, i.e. the commutation of the diagram (see Sec. G.2 for background):

$$
\begin{array}{ccc}
x & \xrightarrow[\quad f(x) - f(x \circ z) \quad]{z} & x \circ z \\
\Phi_{(f,X)} \Big\downarrow & & \Big\downarrow \Phi_{(f,X)} \\
(f * x, X * x) & \xrightarrow[\quad (f*x)(\epsilon) - (f*x)(z) \quad]{z} & T
\end{array}
$$

i.e. the values of $T$ (SE corner) obtained from $x$ (NW corner) via the two paths (NW-SW-SE and NW-NE-SE) are the same.

*Proof.* The value of $T$ via NW-SW-SE is given by $((f*x)*z, (X*x)*z)$, while via NW-NE-SE it is $(f*(x \circ z), X*(x \circ z))$. For any $y \in \mathcal{X}$ we have, by associativity

$$(f*(x \circ z))(y) = f((x \circ z) \circ y) = f(x \circ (z \circ y)) = (f*x)(z \circ y) = ((f*x)*z)(y)$$

Hence $f*(x \circ z) = (f*x)*z$. The proof that $X*(x \circ z) = (X*x)*z$ is identical. Hence expression $T$ has the same value via the two paths.

There are two more things to verify. First that the reward of the N and S transitions are the same: this is obvious since by definition $(f*x)(\epsilon) = f(x \circ \epsilon) = f(x)$ and $(f*x)(z) = f(x \circ z)$. Second, that the conditions for action $z$ to be allowed in the N and S transitions are the same: this is also obvious since the condition in the N transition is $x \circ z \in \bar{X}$ while for the S transition, it is $(X*x)*z \neq \emptyset$ or equivalently $X*(x \circ z) \neq \emptyset$, and we have for any $y \in \mathcal{X}$ (a fortiori for $y = x \circ z$)

$$X * y \neq \emptyset \text{ iff } \exists y' \; y \circ y' \in X \text{ iff } y \in \bar{X}$$

The other commutation, for the null action, is obvious:

$$
\begin{array}{ccc}
x & \xrightarrow[0]{\epsilon} & x \\
\Phi_{(f,X)} \Big\downarrow & & \Big\downarrow \Phi_{(f,X)} \\
(f * x, X * x) & \xrightarrow[0]{\epsilon} & (f * x, X * x)
\end{array}
$$

since the condition for $\epsilon$ to be allowed in the N transition is $x \in X$ and in the S transition is $\epsilon \in X*x$, and the two conditions are equivalent by definition. $\qquad \square$

Observe that the core of the proof is that $*$ defines an action of monoid $\mathcal{X}$ on the right of $\mathcal{F}_{\mathcal{X}}$, forming a kind of flow (in the mathematical sense), where the continuous monoid $(\mathbb{R}, +, 0)$ in the usual definition of flow is replaced by the discrete $(\mathcal{X}, \circ, \epsilon)$. Using Prop. 3, the above result can be strictly equivalently reformulated as:

Let $\boldsymbol{\Phi}^{\sqcup} = \bigsqcup_{(f,X) \in \mathcal{F}_{\mathcal{X}}} \boldsymbol{\Phi}_{(f,X)}$ and $\mathcal{M}^{\sqcup} = \bigsqcup_{(f,X) \in \mathcal{F}_{\mathcal{X}}} \mathcal{M}_{(f,X)}$ be the disjoint union of the bisimulation mappings $\boldsymbol{\Phi}_{(f,X)}$ (on their domain side) and that of their corresponding direct MDPs $\mathcal{M}_{(f,X)}$, respectively. Then $\boldsymbol{\Phi}^{\sqcup}$ is a bisimulation $\mathcal{M}^{\sqcup} \leftrightarrow \mathcal{M}$ where $\mathcal{M}$ is the reduced MDP of $\mathcal{X}$.

Now, $\boldsymbol{\Phi}^{\sqcup}$ is obviously surjective, since $\boldsymbol{\Phi}^{\sqcup}(((f, X), \epsilon)) = (f, X)$ for any $(f, X) \in \mathcal{F}_{\mathcal{X}}$. Hence, by Prop. 6, $\mathcal{M}$ is *isomorphic* to the quotient MDP $\mathcal{M}^{\sqcup}/\boldsymbol{\Phi}^{\sqcup}$, hence the name BQ-MDP for Bisimulation Quotiented MDP.

# G  Mathematical background

## G.1  Monoids

Monoids are one of the simplest algebraic structure.

**Definition 2** (Monoid). *A monoid is a triple $(M, \circ, \epsilon)$ where $\circ$ is a binary operation on the set $M$ and $\epsilon$ is a distinguished element of $M$, such that*

$$\forall x, y, z \in M \qquad x \circ (y \circ z) = (x \circ y) \circ z \qquad \textit{(associativity)}$$
$$\forall x \in M \qquad x \circ \epsilon = \epsilon \circ x = x \qquad \textit{(neutral element)}$$

For example the (finite) sequences of elements of an arbitrary set, equipped with concatenation and the empty sequence, forms a monoid. Now, if $M$ is a monoid and $x_{1:n}$ is a sequence of elements of $M$, then the expression $x_1 \circ \cdots \circ x_n$ denotes an element of $M$, independent of the way it is parenthesised. This extends to the case $n=0$ where the expression denotes $\epsilon$. And the mapping $x_{1:n} \mapsto x_1 \circ \cdots \circ x_n$ from the monoid of sequences of elements of $M$ into $M$ is a monoid homomorphism.

A sub-monoid of $M$ is a subset of $M$ which contains $\epsilon$ and is closed under operation $\circ$. Obviously, a sub-monoid of $M$ is itself a monoid. If $S$ is a sub-monoid of $M$, a generator of $S$, if it exists, is a subset $S_o$ of $M \setminus \{\epsilon\}$ such that $S$ is the smallest sub-monoid of $M$ containing $S_o$. It is easy to show that in that case, $S$ is exactly the set of elements of the form $x_1 \circ \cdots \circ x_n$ where $x_{1:n} \in S_o^n$.

## G.2  Bisimulation

Bisimulation is a very broad concept which applies to arbitrary Labelled Transition Systems (LTS). It has been declined in various flavours of LTS, such as Process Calculi, Finite State Automata, Game theory, and of course MDPs (initially deterministic MDPs such as those used here, later extended to stochastic MDPs which we are not concerned with here). We use the following notation to indicate that the transition from state $p$ to state $p'$ with label $\ell$ is valid in LTS $\mathcal{L}$ ($\mathcal{L}$ is omitted when unambiguous).

$$p \xrightarrow[(\mathcal{L})]{\ell} p'$$

Recall that the disjoint union of a family $(S_i)_{i \in I}$ of sets is the set $\bigsqcup_{i \in I} S_i =_{\text{def}} \bigcup_{i \in I} \{i\} \times S_i$. We can define the disjoint union of a family of LTSs as follows:

**Definition 3** (Disjoint union). *If $(\mathcal{L}_i)_{i \in I}$ is a family of LTSs sharing the same label space, each with state space $S_i$, then the* disjoint union $\bigsqcup_{i \in I} \mathcal{L}_i$ *is the LTS $\mathcal{L}$ with state space $\bigsqcup_{i \in I} S_i$ and transitions*

$$(i, p) \xrightarrow[(\mathcal{L})]{\ell} (i, p') \qquad \textit{if} \qquad p \xrightarrow[(\mathcal{L}_i)]{\ell} p'$$

**Definition 4** (Simulation, Bisimulation). *Let $\mathcal{L}_1, \mathcal{L}_2$ be LTSs sharing the same label space and $\mathcal{R}$ a bi-partite relation from the state space of $\mathcal{L}_1$ into that of $\mathcal{L}_2$. $\mathcal{R}$ is a* simulation $\mathcal{L}_1 \to \mathcal{L}_2$ *if*

$$\forall \ell, p, q, p' \text{ s.t. } p\mathcal{R}q, \ p \xrightarrow[(\mathcal{L}_1)]{\ell} p' \quad \exists q' \text{ s.t. } p'\mathcal{R}q', \ q \xrightarrow[(\mathcal{L}_2)]{\ell} q'$$

$\mathcal{R}$ *is a* bisimulation $\mathcal{L}_1 \leftrightarrow \mathcal{L}_2$ *if $\mathcal{R}$ is a simulation $\mathcal{L}_1 \to \mathcal{L}_2$ and $\mathcal{R}^{op}$ is a simulation $\mathcal{L}_2 \to \mathcal{L}_1$.*

Informally, a simulation (resp. bisimulation) is characterised by a commutation property in the following diagram: if the pair of arrows connected to $p$ (resp. to either $p$ or $q$) is valid then so is the "opposite" pair w.r.t. the centre of the diagram.

$$\begin{array}{ccc} p & \xrightarrow{\ \ell\ } & p' \\ \mathcal{R}\big\downarrow & & \big\downarrow\mathcal{R} \\ q & \xrightarrow{\ \ell\ } & q' \end{array}$$

A *homogeneous* simulation (resp. bisimulation) on an LTS $\mathcal{L}$ is a simulation $\mathcal{L}\to\mathcal{L}$ (resp. bisimulation $\mathcal{L}\leftrightarrow\mathcal{L}$). Note that $\mathcal{R}$ is a simulation $\mathcal{L}_1\to\mathcal{L}_2$ (resp. bisimulation $\mathcal{L}_1\leftrightarrow\mathcal{L}_2$) if and only if $\{((1,p),(2,q))|(p,q)\in\mathcal{R}\}$ is a homogeneous simulation (resp. bisimulation) on $\mathcal{L}_1\sqcup\mathcal{L}_2$.

**Proposition 3.** *Let $\mathcal{L}, (\mathcal{L}_i)_{i\in I}$ be LTSs sharing the same label space. For each $i\in I$, let $\mathcal{R}_i$ be a bi-partite relation from the state space of $\mathcal{L}_i$ into that of $\mathcal{L}$, and let $\mathcal{R}^\sqcup=\bigcup_i\{((i,p),q)|(p,q)\in\mathcal{R}_i\}$ be their disjoint union on the domain side. Then $\mathcal{R}^\sqcup$ is a bisimulation $(\bigsqcup_{i\in I}\mathcal{L}_i)\leftrightarrow\mathcal{L}$ if and only if $\mathcal{R}_i$ is a bisimulation $\mathcal{L}_i\leftrightarrow\mathcal{L}$ for each $i\in I$.*

*Proof.* (outline) This is essentially shown by observing that the following two diagrams, where $\mathcal{L}^\sqcup=\bigsqcup_{i\in I}\mathcal{L}_i$, are, by definition, equivalent:

$$\begin{array}{ccc} (i,p) & \xrightarrow[(\mathcal{L}^\sqcup)]{\ \ell\ } & (i,p') \\ \mathcal{R}^\sqcup\big\downarrow & & \big\downarrow\mathcal{R}^\sqcup \\ q & \xrightarrow[(\mathcal{L})]{\ \ell\ } & q' \end{array} \qquad\qquad \begin{array}{ccc} p & \xrightarrow[(\mathcal{L}_i)]{\ \ell\ } & p' \\ \mathcal{R}_i\big\downarrow & & \big\downarrow\mathcal{R}_i \\ q & \xrightarrow[(\mathcal{L})]{\ \ell\ } & q' \end{array}$$

If the commutation property holds for one it holds for the other. $\qquad\qquad\square$

**Proposition 4.** *The identity on the state space of $\mathcal{L}$ is a bisimulation $\mathcal{L}\leftrightarrow\mathcal{L}$. The composition of a bisimulation $\mathcal{L}_1\leftrightarrow\mathcal{L}_2$ and a bisimulation $\mathcal{L}_2\leftrightarrow\mathcal{L}_3$ is a bisimulation $\mathcal{L}_1\leftrightarrow\mathcal{L}_3$. The union of a family of bisimulations $\mathcal{L}_1\leftrightarrow\mathcal{L}_2$ is a bisimulation $\mathcal{L}_1\leftrightarrow\mathcal{L}_2$. The inverse of a bisimulation $\mathcal{L}_1\leftrightarrow\mathcal{L}_2$ is a bisimulation $\mathcal{L}_2\leftrightarrow\mathcal{L}_1$.*

Hence, LTSs with bisimulation form a category in Category theory.

*Proof.* (outline) Let's detail stability by composition, the other cases are similarly obvious. If $\mathcal{R}_1, \mathcal{R}_2$ are the two bisimulations being composed, apply the commutation property to each cell of the following diagram (from top to bottom and vice versa).

$$\begin{array}{ccc} p & \xrightarrow{\ \ell\ } & p' \\ \mathcal{R}_1\big\downarrow & & \big\downarrow\mathcal{R}_1 \\ r & \xrightarrow{\ \ell\ } & r' \\ \mathcal{R}_2\big\downarrow & & \big\downarrow\mathcal{R}_2 \\ q & \xrightarrow{\ \ell\ } & q' \end{array}$$

$\qquad\qquad\square$

As a corollary, observe that the set of homogeneous bisimulations on an LTS $\mathcal{L}$ is stable by reflexive-symmetric-transitive closure. In particular, the union of all bisimulations, called the *bisimilarity* of $\mathcal{L}$, is itself a bisimulation, and it is an equivalence relation.

**Definition 5** (Quotienting). *Given an LTS $\mathcal{L}$ and an equivalence relation $\mathcal{R}$ on its state space, the quotient LTS $\mathcal{L}/\mathcal{R}$ is defined as follows: the label space is the same as that of $\mathcal{L}$; the states are the $\mathcal{R}$-equivalence classes; and the transitions are defined, for any classes $\dot{p}, \dot{p}'$, by*

$$\dot{p}\xrightarrow[(\mathcal{L}/\mathcal{R})]{\ \ell\ }\dot{p}' \quad\textit{if}\quad \forall p\in\dot{p}\ \exists p'\in\dot{p}'\ \ p\xrightarrow[(\mathcal{L})]{\ \ell\ }p'$$

*By extension, if $F$ is a mapping from the state space of $\mathcal{L}$ to an arbitrary set, then $\mathcal{L}/F$ denotes $\mathcal{L}/(F^{op}\circ F)$.*

**Proposition 5.** *Let $\mathcal{R}$ be an equivalence on the state space of $\mathcal{L}$. $\mathcal{R}$ is a (homogeneous) bisimulation on $\mathcal{L}$ if and only if $\in$ is a bisimulation $\mathcal{L}\leftrightarrow\mathcal{L}/\mathcal{R}$.*

*Proof.* We show both implications: first, assume $\mathcal{R}$ is a bisimulation on $\mathcal{L}$.

**1–** Let $p\in\dot{q}$ and $p\xrightarrow{\ell}p'$. Let $q\in\dot{q}$. Hence $p\mathcal{R}q$ and $p\xrightarrow{\ell}p'$. Since $\mathcal{R}$ is a bisimulation, there exists $q'$ such that $q\xrightarrow{\ell}q'$ and $p'\mathcal{R}q'$. Hence for all $q\in\dot{q}$, there exists $q'\in\bar{p}'$ such that $q\xrightarrow{\ell}q'$. Hence by definition $\dot{q}\xrightarrow{\ell}\bar{p}'$ while $p'\in\bar{p}'$.

**2–** Let $p\in\dot{q}$ and $\dot{q}\xrightarrow{\ell}\dot{q}'$. Hence by definition, there exists $p'\in\dot{q}'$ such that $p\xrightarrow{\ell}p'$.

Conversely, assume $\in$ is a bisimulation $\mathcal{L}\leftrightarrow\mathcal{L}/\mathcal{R}$.

**1–** Let $p\mathcal{R}q$ and $p\xrightarrow{\ell}p'$. Hence $p\in\bar{q}$ and $p\xrightarrow{\ell}p'$. Since $\in$ is a bisimulation, there exists $\dot{q}'$ such that $p'\in\dot{q}'$ and $\bar{q}\xrightarrow{\ell}\dot{q}'$. Now $q\in\bar{q}$, hence, by definition, there exists $q'\in\dot{q}'$ such that $q\xrightarrow{\ell}q'$. And $p'\mathcal{R}q'$ since $p',q'\in\dot{q}'$.

**2–** Let $p\mathcal{R}q$ and $q\xrightarrow{\ell}q'$. Hence $q\mathcal{R}p$ and $q\xrightarrow{\ell}q'$, and we are in the previous case up to a permutation of variables. $\square$

**Proposition 6** (Bisimulation Quotienting). *Let $F$ be a mapping from the state space of $\mathcal{L}_1$ to that of $\mathcal{L}_2$. If $F$ is a surjective bisimulation $\mathcal{L}_1\leftrightarrow\mathcal{L}_2$, then there exists a* bijective *(a.k.a.* isomorphic*) bisimulation $(\mathcal{L}_1/F)\leftrightarrow\mathcal{L}_2$.*

*Proof.* $F$ is a bisimulation $\mathcal{L}_1\leftrightarrow\mathcal{L}_2$, hence, by Prop. 4, $F^{op}$ is a bisimulation $\mathcal{L}_2\leftrightarrow\mathcal{L}_1$ and $\mathcal{R}=F^{op}\circ F$ is a bisimulation $\mathcal{L}_1\leftrightarrow\mathcal{L}_1$. Hence, by Prop. 5, $\in$ is a bisimulation $\mathcal{L}_1\leftrightarrow(\mathcal{L}_1/F)$ and, by Prop. 4 again, $\tilde{F}=F\circ\in^{op}$ is a bisimulation $(\mathcal{L}_1/F)\leftrightarrow\mathcal{L}_2$. By construction, $\tilde{F}$ is an injective mapping between the state space of $\mathcal{L}_1/F$ and that of $\mathcal{L}_2$. Indeed

**1–** Consider a class $\dot{p}$ modulo $\mathcal{R}$ and let $p\in\dot{p}$. Hence, by definition, $\dot{p}\tilde{F}q$ where $q=F(p)$. Furthermore, suppose $\dot{p}\tilde{F}q_1$ and $\dot{p}\tilde{F}q_2$. Hence $q_1=F(p_1)$ and $q_2=F(p_2)$ for some $p_1,p_2\in\dot{p}$. Hence, by definition, $p_1\mathcal{R}p_2$ hence $F(p_1)=F(p_2)$, i.e. $q_1=q_2$. Hence $\tilde{F}$ is a mapping.

**2–** Suppose $\dot{p}_1\tilde{F}q$ and $\dot{p}_2\tilde{F}q$. Hence $F(p_1)=F(p_2)=q$ for some $p_1,p_2\in\dot{p}_1\times\dot{p}_2$. Hence, by definition, $p_1\mathcal{R}p_2$, i.e. $\dot{p}_1=\dot{p}_2$. Hence $\tilde{F}$ is injective.

Now, $\in^{op}$ is obviously surjective between the state space of $\mathcal{L}_1/F$ and that of $\mathcal{L}_1$, hence when $F$ is surjective, so is $\tilde{F}$, which is then bijective. $\square$

**Definition 6.** *Given an LTS $\mathcal{L}$, its* transitive closure *is another LTS denoted $\mathcal{L}^*$ on the same state space, where the labels are the finite sequences of labels of $\mathcal{L}$ and the transitions are defined by*

$$p\xrightarrow[(\mathcal{L}^*)]{\ell_{1:n}}p' \quad if \quad \exists p_{0:n} \text{ such that } p=p_0\xrightarrow[(\mathcal{L})]{\ell_1}p_1\cdots\xrightarrow[(\mathcal{L})]{\ell_{n-1}}p_{n-1}\xrightarrow[(\mathcal{L})]{\ell_n}p_n=p'$$

**Proposition 7.** *If $\mathcal{R}$ is a bisimulation $\mathcal{L}_1\leftrightarrow\mathcal{L}_2$, then it is also a bisimulation $\mathcal{L}_1^*\leftrightarrow\mathcal{L}_2^*$.*

*Proof.* (outline) This is essentially shown by successively applying the commutation property to each cell of the following diagram (from left to right):

$$
\begin{array}{ccccccc}
p_0 & \xrightarrow{\ell_1} & p_1 & \dashrightarrow & p_{n-1} & \xrightarrow{\ell_n} & p_n \\
\mathcal{R}\downarrow & {\scriptstyle \ell_1} & \mathcal{R}\downarrow & & \downarrow\mathcal{R} & {\scriptstyle \ell_n} & \downarrow\mathcal{R} \\
q_0 & \xrightarrow{\phantom{\ell_1}} & q_1 & \dashrightarrow & q_{n-1} & \xrightarrow{\phantom{\ell_n}} & q_n
\end{array}
$$

$\square$

**Proposition 8.** *Let $\mathcal{R}$ be an equivalence relation on the state space of $\mathcal{L}$. If $\mathcal{R}$ is a bisimulation on $\mathcal{L}$, then $(\mathcal{L}/\mathcal{R})^*=\mathcal{L}^*/\mathcal{R}$.*

*Proof.* $\mathcal{R}$ is a bisimulation on $\mathcal{L}$, hence $\in$ is a bisimulation $\mathcal{L}\leftrightarrow\mathcal{L}/\mathcal{R}$ (Prop. 6), hence also a bisimulation $\mathcal{L}^*\leftrightarrow(\mathcal{L}/\mathcal{R})^*$ (Prop. 7). Analogously, $\mathcal{R}$ is a bisimulation on $\mathcal{L}$, hence also a bisimulation on $\mathcal{L}^*$ (Prop. 7), hence $\in$ is a bisimulation $\mathcal{L}^*\leftrightarrow\mathcal{L}^*/\mathcal{R}$ (Prop. 5). By composition (Prop. 4) we have $\in\circ\in^{\mathrm{op}}$ is a bisimulation $\mathcal{L}^*/\mathcal{R}\leftrightarrow(\mathcal{L}/\mathcal{R})^*$. But $\in\circ\in^{\mathrm{op}}$ is the identity. Hence $\mathcal{L}^*/\mathcal{R}=(\mathcal{L}/\mathcal{R})^*$. $\square$

# H    Plots of some TSPLib and CVRPLib solutions

Instance pcb442

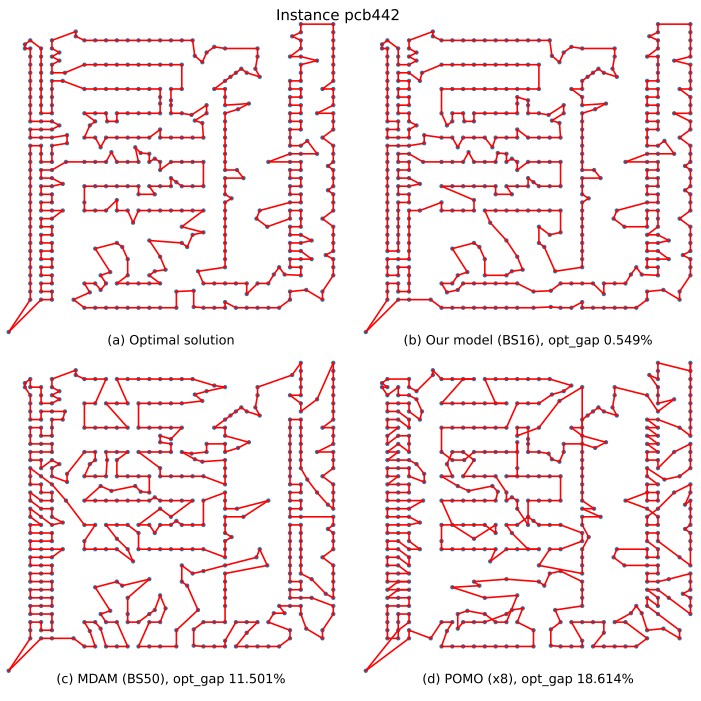

(a) Optimal solution

(b) Our model (BS16), opt_gap 0.549%

(c) MDAM (BS50), opt_gap 11.501%

(d) POMO (x8), opt_gap 18.614%

Instance pr1002

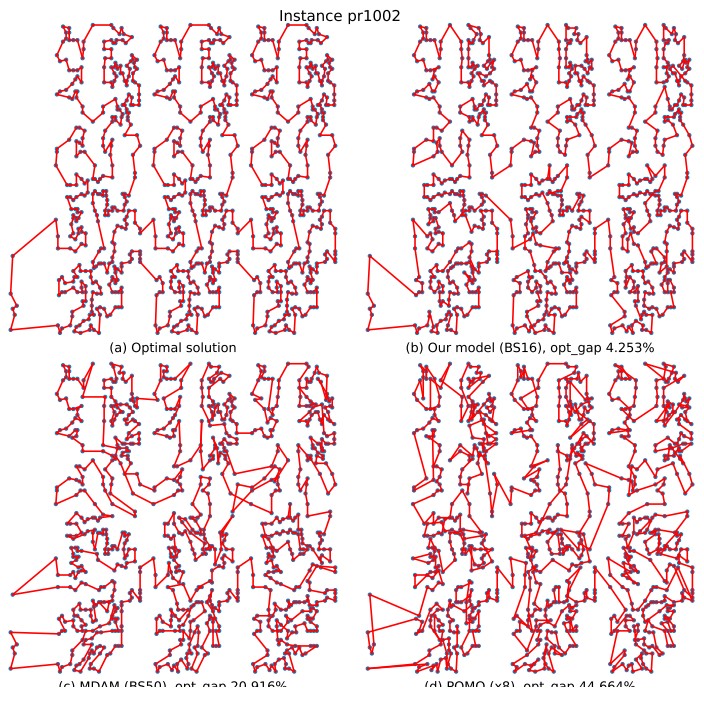

(a) Optimal solution

(b) Our model (BS16), opt_gap 4.253%

(c) MDAM (BS50), opt_gap 20.916%

(d) POMO (x8), opt_gap 44.664%

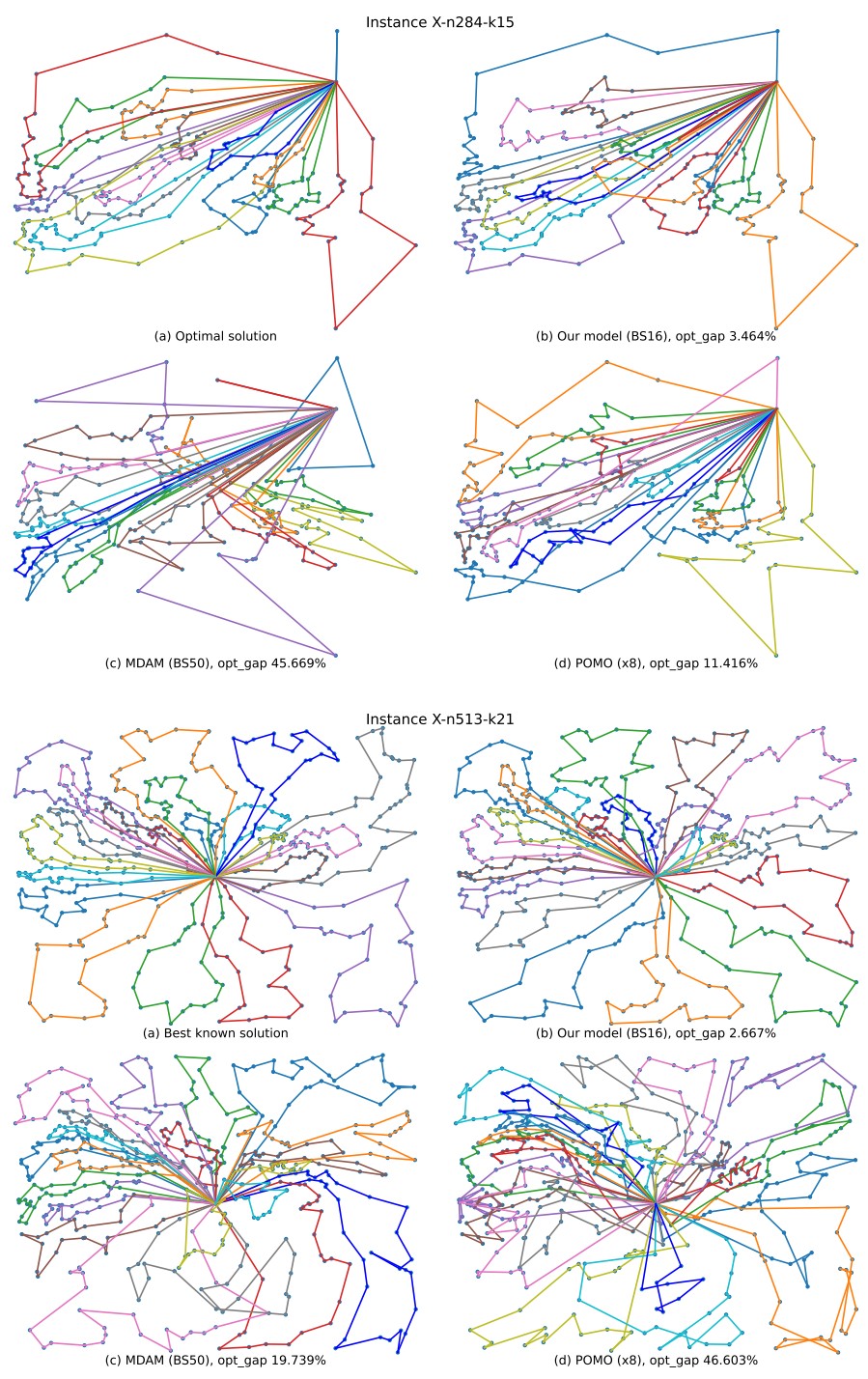

Instance X-n284-k15

(a) Optimal solution

(b) Our model (BS16), opt_gap 3.464%

(c) MDAM (BS50), opt_gap 45.669%

(d) POMO (x8), opt_gap 11.416%

Instance X-n513-k21

(a) Best known solution

(b) Our model (BS16), opt_gap 2.667%

(c) MDAM (BS50), opt_gap 19.739%

(d) POMO (x8), opt_gap 46.603%