# OpenReview forum: "BQ-NCO: Bisimulation Quotienting for Efficient Neural Combinatorial Optimization"
_NeurIPS.cc/2023/Conference — NeurIPS 2023 poster_

### Official Review · Reviewer_e3VL · 2023-07-01

**Soundness:** 2 fair
**Presentation:** 2 fair
**Contribution:** 2 fair
**Rating:** 5
**Confidence:** 4

**Summary:**

Based on Bisimulation Quotienting (BQ), this paper proposed a new MDP formulation named BQ-MDP, to better formalise the process of solution construction in neural constructive solvers for CO problems. The proposed BQ-MDP reduces the state space and is designed to better reflect the symmetry of sub-problems encountered during solution construction. A new deep model named BQ-Transformer (based on Transformer) is designed according to the BQ-MDP, which is leveraged to solve CO problems including TSP, CVRP, and KP. The deep model is trained via supervised learning and the results showed better generalization performance of the learned model across problem sizes and distributions compared to several baselines.

**Strengths:**

- The introduction of BQ into MDP formulations for neural constructive solvers is novel and interesting.
- While several works like POMO (NeurIPS-20) [1], Sym-NCO (NeurIPS-22) [2] and Pointformer (AAAI-23) [3] have tried to explore the symmetries of CO problems, the idea of exploiting the symmetries during the recursive construction process is novel.
- The proposed model showed impressive generalization performance across size and distribution, outperforming several baselines.
- The exploration of BQ-PerceiverG to reduce computational complexity.


**Weaknesses:**


- The clarity of the paper, particularly the mathematical formulations in Section 2.1 and Section 3.1, is limited. Examples include but are not limited to: the concept of the "step set Z" being “a set of partial solutions obtained by one construction step” is unclear, as it is not explicitly stated whether this step is based on an empty solution or an existing partial solution; It is said that “if x, y are partial solutions, x◦y denotes the result of applying the sequence of construction steps yielding x followed by that yielding y”, but I did not follow this definition. If x = z_1◦ · · · ◦ z_n,  does it means that z_i are partial solutions?

- The novelty (about the model architecture) may be limited. The proposed architecture seems to share similarities to an earlier exploration in [4] which also suggests re-encoding the sub-problems during solution construction. Furthermore, in the model designs, the claim about the elimination of the autoregressive decoder may not be compelling. Essentially, the proposed model, similar to AM and POMO, also requires multiple calls to select nodes for solution construction. And the proposed model still follows the encoder-decoder structure, as the last linear layer (used for generating the probability of selecting nodes via softmax) is the decoder.

- The model is trained using imitation learning, which necessitates labeled data and pre-existing solvers to generate high-quality solutions. It is thus not clear if the credits for better performance should be given to the imitation learning/labeled data, or the proposed new architecture.

- The experimental comparisons raise several concerns which make the advantages of the proposed method unclear:
1. On TSP-100 and CVRP-100 (without generalization), the BQ-Transformer performs worse than POMO despite using a longer runtime, particularly on CVRP. The BQ-Transformer seems to struggle with handling VRP constraints effectively.
2. Furthermore, when BQ-Transformer is equipped with beam search, there is no comparison made with POMO under similar settings, e.g., the POMO + beam search as proposed in the SGBS paper (NeurIPS2022) [5].
3. Are the test datasets the same as DIEMS and DIFUSCO? How many instances are in the test datasets? The reported performance gaps of DIMES and DIFUSCO may be inaccurate, as I could not locate the reported gaps in their papers.
4. Since the models are trained on size 100, in Table 1, it would be clearer to label TSP200, TSP500, and TSP1000 as performance under generalization, rather than mixing them up, which could lead to confusion.
5. While the proposed model exhibits better generalization on larger-scale instances, it is unclear whether the model has the ability to further reduce the objective values (gaps) given a longer time budget, and how good the performance would be when compared to baselines with longer run time.

References:

[1] Kwon, Yeong-Dae, et al. "Pomo: Policy optimization with multiple optima for reinforcement learning." Advances in Neural Information Processing Systems 33 (2020): 21188-21198.

[2] Kim, Minsu, Junyoung Park, and Jinkyoo Park. "Sym-nco: Leveraging symmetricity for neural combinatorial optimization." Advances in Neural Information Processing Systems 35 (2022): 1936-1949.

[3] Jin, Yan, et al. "Pointerformer: Deep Reinforced Multi-Pointer Transformer for the Traveling Salesman Problem." arXiv preprint arXiv:2304.09407 (2023).

[4] Xin, Liang, et al. "Step-wise deep learning models for solving routing problems." IEEE Transactions on Industrial Informatics 17.7 (2020): 4861-4871.

[5]  Choo, Jinho, et al. "Simulation-guided beam search for neural combinatorial optimization." Advances in Neural Information Processing Systems 35 (2022): 8760-8772.

**Questions:**

- Are the test datasets the same as DIEMS and DIFUSCO? How many instances are in the test datasets? Why have the objective values, in addition to the gaps, not been reported in Table 1?

- What are the principal differences between the proposed model architecture and the one in reference [4]? Can we apply [4] to reflect the features of BQ-MDP?

- The claim about the generality of BQ-MDP seems to be overstated. Are there certain Combinatorial Optimization Problems (COPs) that the MDP might struggle to handle? For instance, how would the MDP manage problems where the feasibility of the solution can only be determined when the solution is complete, as in some Simulation Optimization tasks?

- Can the authors clarify the mathematical notations mentioned above? It may be beneficial to include examples (for instance, toy TSP-5 instances) to illustrate the new concepts introduced.

- I could not find the correct Appendix; the current zip file contains the submitted manuscript for ICLR23, not the NeurIPS appendix.


**Limitations:**

The limitation mentioned in the weakness above should be discussed in the revised paper.

---

> ### Author Rebuttal · Authors · 2023-08-09
>
> We thank the reviewer for the detailed feedback. We answer below their questions and clarify important points.
>
> W1/Q4. **"about the step set Z"** We understand the ambiguity and confirm that a step is a partial solution obtained by one construction step *starting from the empty solution*.
>
> W1/Q4. **"About the definition of x◦y and that it may be beneficial to include examples"** It is important to remember that in our definition, *partial solutions* are not linked to an instance but just a CO problem (L79). We provide examples of the introduced concepts in the paragraph "Examples of Solution Spaces" L93. In particular, we mention that for TSP a step is simply a singleton node, a partial solution a sequence of nodes and  x◦y denotes the concatenation of two node sequences.
>
> W1/Q4. **"If $x = z_1◦ · · · ◦ z_n$, does it means that $z_i$ are partial solutions?"** Yes, here $z_i$ are steps and therefore by definition partial solutions.
>
> W2. **"What are the principal differences between the proposed model ... and [4]?"** [4] is specific to TSP/CVRP, where they simply drop visited nodes, while our approach works with any problem amenable to DP. In fact, for CVRP, they need to keep a trace of the visited nodes to restrict the feasible actions; we don't have this issue: it is dealt with by the tail sub-problem mechanism of DP. Please refer to Appendix D.2 for more comparisons. [4] suggest an approximation where part of the encoder is fixed while top layers are recomputed with a mask on already visited nodes. We have tried this approximation in our context and observed a drop in performance, suggesting that solving the precise BQ-MDPs is key to our results.
>
> W2 **"the claim about the elimination of the autoregressive decoder may not be compelling...the proposed model still follows the encoder-decoder structure, as the last linear layer (used for generating the probability of selecting nodes via softmax) is the decoder."**
> While we agree that the last linear layer could be viewed as a decoder, since the whole model is called at each construction step, it seems artificial to us to separate the encoding from the decoding. This is in contrast to AM/POMO where the encoder is called once per instance and then the decoder many times.
>
> W3. **"It is thus not clear if the credits for better performance should be given to the imitation learning/labeled data, or the proposed new architecture."** First, please note that we do not claim proposing a novel architecture. It is a simple transformer model, similar to previous works, with slight adaptations of input and output layers to match the BQ-MDP. The novelty is in the BQ-MDP framework. Also the results are consistent with different architectures (transformer, perceiver and the combination of transformer and GCN as introduced in the General Response above). Experimentally, we show that the combination of efficient imitation training with our models works very well. Note that the BQ-models outperform some other approaches that rely on labeled data (Att-GCN and DIFUSCO).
>
> **"when BQ-Transformer is equipped with beam search, there is no comparison made with POMO under similar settings, e.g., the POMO + beam search as proposed in the SGBS paper [5]"**. We added this baseline in the Table in the rebuttal document. The BQ-transformer + beam search outperforms SGBD on 7 out of 8 datasets, e.g. with a reduction of the opt gap from 25% (in 6h) to 1.38% (in 38min) on TSP1000.
>
> **"C3/Q1. Are the test datasets the same as DIEMS and DIFUSCO? How many instances are in the test datasets? The reported performance gaps of DIMES and DIFUSCO may be inaccurate, as I could not locate the reported gaps in their papers."** Yes same datasets (introduced in Fu et al 2021). They contain 10k instances for TSP100 and 128 for TSP200/500/1000. The reported results in our Table 1 correspond to the "interpolation results" presented in Table 5 of the DIMES paper and Figure 3 in DIFUSCO paper. The objective values are not reported to save space and because find the optimality gap to be the most informative. However we can add them in the Appendix.
>
> C5. **"While the proposed model exhibits better generalization on larger-scale instances, it is unclear whether the model has the ability to further reduce the objective values (gaps) given a longer time budget and how good the performance would be when compared to baselines with longer run time."** Given a longer time budget, one could use a search procedure or improvement heuristic on top of our model. We do this already with a simple beam search (see "BQ-Transformer bs16" in Tables 1,2) and obtain a significant performance gain at the cost of a somewhat reasonable increase of computing time (from less than 2 min to at most 38 min). With the beam-search, our model outperforms the baselines for TSP, CVRP, ATSP, OP and KP on the vast majority of the test datasets, outperforming baselines that use 2-opt (DIFUSCO) or MCTS (Att-GCN). Tests with even longer running times and more sophisticated search methods such as Active Search are possible but outside the scope of this paper.
>
> Q3. **"Are there certain Combinatorial Optimization Problems (COPs) that the MDP might struggle to handle? For instance, how would the MDP manage problems where the feasibility of the solution can only be determined when the solution is complete.."** As stated L64: "We present a generic and principled framework to derive a direct MDP given any COP with minimal requirements". And indeed we make the assumption (L76) that the feasibility of the solution can easily be ensured during the construction process. This is a necessary condition for constructive approaches to work and it is not specific to the BQ-MDP.
>
> Q5. **"I could not find the correct Appendix; the current zip file contains the submitted manuscript for ICLR23, not the NeurIPS appendix."**
> We double-checked and confirm that the appendix is accessible through the supplementary material link in the submission page.

---

> > ### Comment · Reviewer_e3VL · 2023-08-12
> > **Thanks for the response**
> >
> > I appreciate the authors' detailed response which addressed some of my concerns, and I am now able to access the correct Appendix. I will increase my score, however, I still have some concerns.
> >
> > 1. Performance Concerns:
> > * 1.1 While achieving better cross-size generalization performance (train on size 100 and test on larger size), the in-size performance (train and test on size 100) of the approach is limited (even worse than POMO on TSP and CVRP).
> > * 1.2 The added results in PDF compare with a basic version of SGBS (4,4), overlooking the superior EAS+SGBS variant where the efficient active search (EAS) is the design that aims to boost the generalization of POMO on larger scales.
> > * 1.3 Following the above, it is thus interesting to evaluable how good the performance of the BQ-MDP is when compared to other recently developed frameworks that aim to boost the cross size/distribution performance of a neural solver (shall be discussed and compared). In other words, it is hard to conclude that BQ-MDP is the SOTA framework for generalization enhancement (no comparison in the paper).
> >
> > 2. Use of Imitation Learning:
> > * 2.1 The approach relies on imitation learning and labeled data, diverging from the original motivation of constructive NCO solvers like AM and POMO which aim to learn solvers without much human knowledge.
> > * 2.2 While the proposed approach used imitation learning, key baselines like POMO and reference [4] were RL-based. It is unclear if the improved generalization performance is attributed to imitation learning or the BQ-MDP framework. What if we train the baselines (e.g. the architecture in ref[4] or POMO) using high-quality solutions generated in this paper instead of RL?
> >
> > 3. Presentation:
> > More clarity is needed around the BQ-MDP framework's details, what steps are needed to adapt it to various architectures/existing approaches (as a framework), and the details of mathematical formulations.

---

> > > ### Author Response · Authors · 2023-08-14
> > > **Response to concerns (1/2)**
> > >
> > > We thank the reviewer for acknowledging our response and recapitulating their concerns, which we address below:
> > >
> > > **1.1 better cross-size generalization performance but limited in-size performance (worse than POMO on TSP100 and CVRP100).**
> > >
> > > We would like to highlight that we have tested our approach on 5 problems and 35 benchmarks in total (cf the Rebuttal document Table, Table 2 in the paper and 3 in the Appendix). Our models yield the best performance among the neural methods on 30 out of these 35 testsets. When they are not the best, their optimality gaps are still very good: less than 1%. It seems unrealistic to expect one model to perform better than all the baselines for this many problems/datasets.
> > >
> > > **1.2 Comparison with SGBS (4,4), but overlooking the superior EAS+SGBS  where the efficient active search (EAS) is the design that aims to boost the generalization of POMO on larger scales.**
> > >
> > > Looking at the reported results for EAS+SGBS in the original paper, this method does not seem to clearly outperform our models in terms of generalization:
> > >
> > > * On TSP200, BQ-transformer+bs gets 0.09% in 3 minutes for 128 instances while EAS+SGBS gets 0.196% in 30 hours for 1k instances (therefore at least 3 hours for 128 instances).
> > > * On CVRP200, BQ-transformer+bs gets 0.77% in 3 minutes for 128 instances while EAS+SGBS gets 0.40% in 50 hours for 1k instances (therefore at least 6 hours for 128).
> > >
> > > More importantly, the combination of Efficient Active Search and SGBS is specifically designed for longer running times. To cite the paper: "SGBS can be further partnered with efficient active search (EAS) [1] to achieve even better performance over longer time spans." Indeed the best performances reported are for tens of hours of computation. This is valuable to provide the best possible solutions in settings where computation time is not so critical. Our method on the other hand is designed to provide an efficient heuristic: i.e. good quality solutions, fast. We compare it to methods in that category. Pairing our models with EAS, MCTS, or k-opt to improve the solution quality by exploiting longer running times is interesting but outside the scope of this paper.
> > >
> > > **1.3 Missing comparison with other recently developed frameworks that aim to boost the cross size/distribution performance of a neural solver.**
> > >
> > > Regarding the neural baselines, we have compared to the strongest and most recent constructive approaches that we are aware of, including: AttGCN (AAAI 2021), DIMES (Neurips 2022), SymNCO (Neurips 2022), MatNet (Neurips 2022), SGBS (Neurips 2022), DIFUSCO (ArXiv 2023). Most of these were specifically designed to address the scalability challenge of neural combinatorial optimization. Except EAS+SGBS which addresses another setting (see previous point), we would appreciate if the reviewer could point out which recently developed frameworks they are referring to.
> > >
> > > **2.1 The approach relies on imitation learning and labeled data, diverging from the original motivation of constructive NCO solvers like AM and POMO which aim to learn solvers without much human knowledge.**
> > >
> > > Indeed we have a different motivation, which is to exploit that even for NP-complete problems, it is generally possible to efficiently get good-quality solutions up to a certain size -- using hand-crafted heuristics or generic Integer Programming solvers. This is also the motivation behind approaches such as AttGCN for TSP or recently DIFUSCO. We have demonstrated that this is possible with all 5 different problems.
> > >
> > > **2.2 While the proposed approach used imitation learning, key baselines like POMO and reference [4] were RL-based. It is unclear if the improved generalization performance is attributed to imitation learning or the BQ-MDP framework.**
> > >
> > > We show that our models trained by imitation perform very well -- better than some models trained in RL (e.g. AM/POMO and [4]) and other models trained in a supervised way (AttGCN, DIFUSCO). Imitation learning by itself is not known to be particularly efficient at generalization, on the contrary (see eg. the "generalization gap" of IL in robotics manipulation [1], where it is heavily used), and we see no reason to believe that it is the case here.
> > >
> > > [1] Xie et al, Decomposing the Generalization Gap in Imitation Learning for Visual Robotic Manipulation, arXiv 2023.
> > >
> > > **What if we train the baselines (e.g. the architecture in ref [4] or POMO) using high-quality solutions generated in this paper instead of RL?**
> > >
> > > This is an interesting question that is triggered by our work. Future works may revisit those models with imitation learning.

---

> > > ### Author Response · Authors · 2023-08-14
> > > **Response to concerns (2/2)**
> > >
> > > **Presentation:**
> > >
> > > **3.1. What steps are needed to adapt the BQ-MDP framework to various architectures/existing approaches**
> > >
> > > If one wanted to test our framework with another architecture they would need to: (i) adapt the input and output layers of the given architecture to handle the BQ-MDPs (the input is an instance, the output a step, typically a node); (ii) implement the transition function: given an instance and a step, compute the new instance which corresponds to the tail subproblem after applying the construction step; (iii) then train the new architecture .
> > >
> > > The updated manuscript will contain an illustration of this process for 5 problems and 3 architectures (transformer, perceiver, transformer+GCN). We have also provided our code for the reviewing process and plan to open-source it for easier adoption.
> > >
> > > **3.2. More clarity around details of mathematical formulations**
> > >
> > > Beyond the steps and $\circ$ operator that we have addressed above, we would be grateful if the reviewer could point out which details of the mathematical formulation remain unclear.

---

> > > > ### Comment · Reviewer_e3VL · 2023-08-15
> > > > **Thanks for the reply for my further comment**
> > > >
> > > > I want to thank the authors for the detailed reply. Please revise the paper according to our discussions. Given the promising performance and the setting of finding solutions in a fast speed, I will increase my score to 5. Nevertheless, I would still suggest the authors to consider the following baseline frameworks that boost the cross size/distribution performance of a neural solver.  I have no further questions.
> > > >
> > > > * Learning to Delegate for Large-scale Vehicle Routing (NeuIPS 2021)
> > > > * Learning Generalizable Models for Vehicle Routing Problems via Knowledge Distillation (NeurIPS 2022)
> > > > * RBG: Hierarchically Solving Large-Scale Routing Problems in Logistic Systems via Reinforcement Learning (KDD 2023)
> > > > * H-TSP: Hierarchically Solving the Large-Scale Travelling Salesman Problem (AAAI 2023)
> > > > * Select and Optimize: Learning to solve large-scale TSP instances (AISTATS 2023)
> > > > * Generalize Learned Heuristics to Solve Large-scale Vehicle Routing Problems in Real-time (ICLR 2023)
> > > > * Meta-SAGE:Scale Meta-Learning Scheduled Adaptation with Guided Exploration for Mitigating Scale Shift on Combinatorial Optimization (ICML 2023)
> > > > * Towards Omni-generalizable Neural Methods for Vehicle Routing Problems (ICML 2023)

---

> > > > > ### Author Response · Authors · 2023-08-19
> > > > > **Thanks for the additionnal references**
> > > > >
> > > > > We thank the reviewer for their response and for listing these references. We want to highlight that we propose **end-to-end models**, trained on instances with 100 nodes that (0-shot) generalize to instances with up to 4000 nodes. We compare our models to state-of-the-art end-to-end neural models (some with an additional search method). Most of the listed references, despite treating generalization, do not fall into this category. We agree that they should be mentioned in the related works, however they do not change our claim about the state-of-the-art generalization performance of our models. It is also worth noting that most of these references are specifically designed for one problem whereas we propose a generic approach and successfully apply it to five different problems.
> > > > >
> > > > > Essentially:
> > > > > * [1], [3] and [6] are VRP-specific **hybrid** approaches that are based on the classic devide-and-conquer principle: they learn how to split large instances into smaller ones, that are then efficiently solved by external specialized solvers (LKH or HGS).
> > > > > * [2] is limited to TSP/CVRP instances with up to 100 nodes and mentions generalization lo larger instances as future work;
> > > > > * [4] is TSP-specific and does not outperform Att-GCG+MCTS, which we have as a baseline;
> > > > > * [5] is TSP-specific and a quick comparison with their Table 2 shows that our model outperforms theirs on TSP200/500/1000.
> > > > > * [7] and [8] propose meta-learning frameworks that allow the adaptation of a pretrained model and therefore could be combined with BQ-NCO as base model. Both are ICML papers, published after the NeurIPS deadline, and would be considered concurrent works.
> > > > >
> > > > > We are happy to discuss these in more detail in the updated related works section. We sincerely thank the reviewer again for taking the time to thoroughly discuss our work.
> > > > >
> > > > > [1] Learning to Delegate for Large-scale Vehicle Routing (NeuIPS 2021)
> > > > > [2] Learning Generalizable Models for Vehicle Routing Problems via Knowledge Distillation (NeurIPS 2022)
> > > > > [3] RBG: Hierarchically Solving Large-Scale Routing Problems in Logistic Systems via Reinforcement Learning (KDD 2023)
> > > > > [4] H-TSP: Hierarchically Solving the Large-Scale Travelling Salesman Problem (AAAI 2023)
> > > > > [5] Select and Optimize: Learning to solve large-scale TSP instances (AISTATS 2023)
> > > > > [6] Generalize Learned Heuristics to Solve Large-scale Vehicle Routing Problems in Real-time (ICLR 2023)
> > > > > [7] Meta-SAGE:Scale Meta-Learning Scheduled Adaptation with Guided Exploration for Mitigating Scale Shift on Combinatorial Optimization (ICML 2023)
> > > > > [8] Towards Omni-generalizable Neural Methods for Vehicle Routing Problems (ICML 2023)

---

### Official Review · Reviewer_8qf4 · 2023-07-04

**Soundness:** 3 good
**Presentation:** 3 good
**Contribution:** 3 good
**Rating:** 6
**Confidence:** 4

**Summary:**

The paper proposes to use the notion of bisimulation quotienting (BQ) to define state spaces with the minimal information that is required to solve a combinatorial optimization (sub)problem. This results in a simpler architecture for the model used in training. Empirically, the methods advance state-of-the-art performance on a couple of routing problems.


**Strengths:**

The paper rigorously formulates BQ-MDP for CO problems. I find this method applicable to a wide range of CO problems whenever it could be formulated as dynamic programming.

The empirical results show that the method can generalize to larger instances than training.

The paper includes many baselines, including some recent ones e.g. DIFFUSCO.

I also like the idea of testing the method with a lightweight version of transformers.


**Weaknesses:**

This method is limited to problems where finding feasible solutions are easy since it is mainly a construction heuristic.  Other problems like 3-SAT  (decision version of the Satisfiability problem, as opposed to MAX-SAT) that have a much constrained solution space might not be suitable for this method, though the BQ-MDP formulation is still applicable.

For people who are familiar with dynamic programming (DP), the idea behind the approach is pretty straightforward — In DP, it is crucial to identify the minimal state representation to solve a subproblem to reduce complexity. This paper formulates this idea in a more mathematical fashion.

I find this method applicable to a wide range of CO problems whenever it could be formulated as dynamic programming. But the authors didn’t emphasize this enough — it simply states it as a generic framework for COP. It is not obvious to readers how widely this method is applicable to COP.


In experiments, it would be nice to also report variances. The benchmarks used are mostly in euclidean space, which gives the readers an impression that this might only work for this case. Are there non-euclidean routing problem benchmark that you could test your approach on?

It seems that the method is not restricted to a particular learning algorithm or architecture. Imitation learning is just a way to go, but it wasn’t clear in the paper. In other words, would other learning methods like RL also be applicable?

**Questions:**

See the weaknesses section.

This paper seems relevant to this work: https://arxiv.org/pdf/2006.01610.pdf
Might be good to discuss its relevance in the paper.

Is the method more sample efficient that the other works? Intuitively, this should be one of the benefits because you are reducing states that are essentially the same but with different representations of partial solutions.

**Limitations:**

I don't see any limitations other than those pointed out already.

---

> ### Author Rebuttal · Authors · 2023-08-09
>
> We thank the reviewer for the insightful and detailed feedback. We address their concerns and questions below.
>
> W1. **"This method is limited to problems where finding feasible solutions are easy since it is mainly a construction heuristic. Other problems like 3-SAT ... that have a much constrained solution space might not be suitable for this method, though the BQ-MDP formulation is still applicable."** Indeed we assume that feasibility of a final solution can be ensured through conditions on the partial solutions at each step of the construction process (L72 and L121). We argue that it is a necessary condition for all constructive approaches, although it is often not mentioned explicitly. This assumption may not hold for problems with complex, global constraints. In that case, the MDP may have dead-end states, which not all training algorithms are capable of dealing with (ref [23] on L124). We plan to make this assumption more explicit in the introduction.
>
> W2. **"For people who are familiar with dynamic programming (DP), the idea behind the approach is pretty straightforward — In DP, it is crucial to identify the minimal state representation to solve a subproblem to reduce complexity. This paper formulates this idea in a more mathematical fashion."** Although we agree that the idea of restricting the state to the most compact representation may be intuitive, we believe that the mathematical formulation allows to prove the soundness of the approach and points out its requirements — e.g. the feasibility assumption (L121) and the need of a uniform instance parametrization (Sec 3.3).
>
> W3. **"I find this method applicable to a wide range of CO problems whenever it could be formulated as dynamic programming. But the authors didn’t emphasize this enough — it simply states it as a generic framework for COP. It is not obvious to readers how widely this method is applicable to COP."**
>
> We claim that the direct MDP framework (Sec 2) is generic and could be applied to any COP. The BQ-MDP is also generic in its abstract form (Sec 3.2). However, in order to implement the BQ-MDP, the optimality principle of dynamic programming is needed (Sec 3.3). We plan to emphasis this more in the introduction.
>
> W4. **"In experiments, it would be nice to also report variances."**
> In our experience, the variances are pretty small. Similarly to all previous works on the same benchmarks (e.g. AM, POMO, MDAM, DIMES, DIFUSCO etc) we did not report them in the table because of lack of space. However we could add them in the Appendix.
>
> W5. **"The benchmarks used are mostly in euclidean space, which gives the readers an impression that this might only work for this case. Are there non-euclidean routing problem benchmark that you could test your approach on?"**
>
> Yes, non-Euclidian (actually not even symmetric) routing results have been added. See the general response and the pdf attached to the rebuttal.
>
> W6. **"It seems that the method is not restricted to a particular learning algorithm or architecture. Imitation learning is just a way to go, but it wasn’t clear in the paper. In other words, would other learning methods like RL also be applicable?"**
>
> Absolutely. Indeed the BQ-MDP, formulated as a bisimulation of the direct MDP, is not tied to a training procedure and therefore could be solved with RL. We mention this in the conclusion (L382) but could mention it earlier in the paper.
>
> Q1. **"Discussion of https://arxiv.org/pdf/2006.01610.pdf"**
>
> Our paper focuses exclusively on MDP-based solutions to COP, while admittedly there are many more approaches, including the hybrid CP-DRL approach proposed in this reference. Thanks to its CP component, this approaches may handle COP with more challenging feasibility constraints. It would make sense to mention it indeed in the paper as an alternative for those problems.
>
> Q2. **"Is the method more sample efficient that the other works? Intuitively, this should be one of the benefits because you are reducing states that are essentially the same but with different representations of partial solutions."**
>
> We use 1M training instances per problems (TSP100, CVRP100 and KP100). Our training is definitely more sample efficient than RL-based approaches (e.g. AM/POMO training uses 128M/20M TSP100 instances) but this is certainly due to the supervision. When we compare to other SL-based methods, we used the same training datasets as Joshi et al 2019/DPDP of 1M instances while DIFUSCO used about 1.5M instances for TSP100. However the sample efficiency is more obvious at a higher level since the generalization ability of our model allows us to only train one model per problem while all the models above are trained separately for each instance size.

---

> > ### Comment · Reviewer_8qf4 · 2023-08-12
> > **Thanks for the response**
> >
> > I want to thank the author for the detailed responses. They have mostly resolved my concerns. I encourage the authors to include the details clarified during the rebuttal in the main paper.
> >
> > I have also read the other reviews and remain supportive of acceptance.

---

> > > ### Author Response · Authors · 2023-08-14
> > > **Thanks for the feedback**
> > >
> > > We thank the reviewer for acknowledging our responses and for taking the time to read the other reviews. We are pleased that the reviewer supports the acceptance and confirm that we will include the additional clarifications as well as the new benchmarks in the main paper (using the additional page to accommodate the new content).

---

### Official Review · Reviewer_cBr6 · 2023-07-06

**Soundness:** 3 good
**Presentation:** 3 good
**Contribution:** 3 good
**Rating:** 6
**Confidence:** 4

**Summary:**

The paper presents a generic framework for deriving direct MDPs for a given combinatorial optimization problem (COP) and propose a method for reducing the MDP using symmetry-focused bisimulation quotienting. Specifically, the paper presents a generic bisimulation that can be applied to a wide range of recursive problems. The paper presents an instantiation of the approach to three well known problems: TSP, CVRP, and Knapsack using a simple Transformer-based architecture with minor adaptation per application. The experiments show that the proposed approach obtains significantly better performance in generalizing to larger problems.

**Strengths:**

- Novel and interesting approach to enhance neural combinatorial optimization problem by exploiting the recursive nature of many COPs.
- The proposed technical approach seems to be relatively generic and can potentially be applied to a range of combinatorial optimization problems.
- Experiments show significant improvement in generalizing to larger problems compared to the baselines.

**Weaknesses:**

- Experiments in the paper are limited to three benchmarks domains. While these are important and somewhat diverse domains, previous work has also considered combinatorial optimization problems like maximum cut, maximum independent set, etc. Further, as TSP and CVRP are highly related and the results for KP show somewhat different performance patterns, it is not clear how well this approach works beyond the limited set of benchmarks. Specifically, the results on KP that are only presented in the appendix show more limited improvement over POMO compared to TSP and CVRP (no other baselines). Taking into account that POMO is a relatively poor baseline on TSP/CVRP in terms of generalizing to larger instances, this raises the concern that the proposed method may not perform as well on domains beyond TSP/CVRP.

 - Comparison with previous symmetry-related enhancements such as Sym-NCO is missing. While the authors note that such approaches are "orthogonal to our approach", it is still important to evaluate the benefit of the proposed approach in presence of these augmentations.

- The approach still requires optimal solutions for training. Such optimal solutions are typically intractable to compute beyond certain problem size. While the proposed approach shows very good performance in generalizing to larger problems, there may be a limit to the scalability (the paper presents results for up to one order of magnitude larger test problems). It is interesting to see if this can be paired with RL fine-tuning to further improve performance on larger instances.

**Questions:**

I would appreciate the authors' response to my main concerns listed above.

**Limitations:**

There is no explicit discussion of limitation, however I have no concerns about the work.

---

> ### Author Rebuttal · Authors · 2023-08-09
>
> We thank the reviewer for the thoughtful and constructive feedback. It inspired us to make more experiments that we believe make the paper stronger. We address here the main concerns:
>
> W1. **"Experiments in the paper are limited to three benchmarks domains..."**
> New experiments with the Orienteering Problem and Asymmetric TSP have been added. See general response above and pdf attached to rebuttal.
>
> W2. **"Comparison with previous symmetry-related enhancements such as Sym-NCO is missing..."**
> Thank you for pointing out this baseline. Comparison with Sym-NCO has been added. See general response above and pdf attached to rebuttal.
>
> W3. **"The approach still requires optimal solutions for training... It is interesting to see if this can be paired with RL fine-tuning to further improve performance on larger instances."**
> We currently solve instances with up to 4000 nodes (TSPlib). RL-finetuning would indeed be interesting to explore. We note that current neural models that achieve a good performance on even larger graphs such as TSP10,000 rely on improvement heuristics on top of the neural model (MCTS or 2-opt for AttGCRN, DIMES, DIFUSCO), which would be another possibility for future work.

---

> > ### Comment · Reviewer_cBr6 · 2023-08-17
> > **Thank you for your response**
> >
> > I thank the authors for their response and the additional results. My main concerns have been addressed. I remain supportive of accepting the paper.

---

### Official Review · Reviewer_fu7F · 2023-07-06

**Soundness:** 3 good
**Presentation:** 2 fair
**Contribution:** 3 good
**Rating:** 6
**Confidence:** 4

**Summary:**

This paper first present a formulation of Combinatorial Optimization Problems (COPs) as Markov Decision Processes (MDPs) that can leverage symmetries of COPs to improve out-of-distribution robustness. Based on the MDP formulation, the authors then introduce the Bisimulation Quotienting to the MDP, namely BQ-MDP, which can reduce the state space. Finally, the authors introduce a simple attention-based policy network to learn policies in the BQ-MDP. Experiments demonstrate that the proposed method achieves the state-of-the-art performance on both  synthetic and realistic benchmarks.

**Strengths:**

1. To the best of my knowledge, the idea of introducing the Bisimulation Quotienting (BQ) to exploit symmetries of COPs is novel.
2. Experiments demonstrate that the proposed method achieves the state-of-the-art performance on both  synthetic and realistic benchmarks. Moreover, the method achieves impressive generalization performance to much larger instances than seen during training.


**Weaknesses:**

The presentation could be improved.
1. The authors claim that their formulation is generic for any CO problems. However, the formulation seems specific for the binary integer linear programming problems. Thus, it would be more convincing if the authors could explain how the proposed formulation applies to production planning problems with both continuous and discrete variables.
2. The details about the proposed method are unclear to me. For example, the authors may want to explain why and how introducing Bisimulation Quotiented can exploit the symmetries of COPs. Moreover, the details about the impact of the BQ-MDP on model architecture is unclear.


**Questions:**

Please refer to Weaknesses for my questions.

**Limitations:**

Yes

---

> ### Author Rebuttal · Authors · 2023-08-09
>
> We thank the reviewer for the insightful feedback and respond below to their questions and will be happy to further clarify any point.
>
> W1. **"The formulation seems specific for the binary integer linear programming problems. Thus, it would be more convincing if the authors could explain how the proposed formulation applies to production planning problems with both continuous and discrete variables."**
>
> The proposed formulation only applies to "purely" combinatorial problems, i.e. where each instance has a finite set of feasible solutions (as stated L72). This assumption is essential, and is repeatedly used in the proofs of all the properties (Appendix E). It would apply to Integer Linear (or non-linear) programs which are equivalent to binary integer programs. An extension of the framework to problems with both continuous and discrete variables, although outside of the scope of the paper, would be interesting to explore as future work.
>
> W2. **"the authors may want to explain why and how introducing Bisimulation Quotiented can exploit the symmetries of COPs."**
>
> The symmetry that we seek to exploit is that many pairs of an instance and a partial solution lead to the same tail subproblem and therefore an optimal policy should produce the same action at such pairs. Bisimulation quotienting allows to reduce the state space from (instance - partial solution) pairs to tail subproblems while preserving the equivalence between the MDP solving and the original COP solving.
>
> W2. **"the details about the impact of the BQ-MDP on model architecture is unclear."**
>
> As model architecture we use a transformer, similar to previous work. When dealing with states that are (instance - partial solution) pairs, previous works usually start by encoding the instance once and then mimic the autoregressive construction of the partial solution in decoding. With BQ-MDP, the input of the model is an instance and the output is one action leading to a new instance (tail subproblem). Thus we get rid of the autoregressive decoder, but we call the model at each step of the construction process, with the tail subproblem as input.

---

> > ### Comment · Reviewer_fu7F · 2023-08-16
> > **Thanks for the response**
> >
> > Thanks for the response to address most of my concerns. I will keep my supportive score for this paper unchanged.

---

### Author Rebuttal · Authors · 2023-08-09

We thank the reviewers for their insightful feedback. We are encouraged that they appreciated the **novelty of the BQ-MDPs** (fu7F,cBr6,e3VL), the **wide applicability of our approach** (cBr6,8qf4) and its **impressive generalization results** (fu7F,e3VL) with the simple transformer model as well as its **efficiency with the lightweight perceiver** architecture (8qf4,e3VL).

A **common concern was the limited set of benchmark problems**, questioning whether the excellent generalization performance would hold beyond the tested problems. Indeed cBr6 noted the limitation to 3 problems, 8qf4 the limitation of TSP/CVRP to the Euclidian setting and e3VL mentioned that our model may struggle with the CVRP constraints.

To address these concerns, we have applied our approach to **two more problems**: the **Orienteering Problem (OP)** (which has another kind of constraints) and the **Asymmetric TSP (Asym-TSP)** (non-Euclidian routing problem). We present in the rebuttal PDF file a new table with:

* the original paper results (for TSP, CVRP, KP) with 2 additionnal baselines: Sym-NCO [1] (suggested by cBr6) and SGBS [2] (suggested by e3VL).
* the new results for the OP and Asym-TSP with the appropriate neural baselines and classic solvers (see Details below).

The results for OP and ATSP are consistent with those of TSP, CVRP and KP: the greedy rollout of the BQ models already gives a very good generalization performance; with a beam search (with beam width of 16), our model outperforms the neural baselines for 7 out of 8 test datasets. Notably, BQ-GCN-transformer is the first model, to the best of our knowledge, that is able to solve ATSP instances of sizes 500 and 1000. And for OP100 and 200, the BQ-transformer model outperforms even the state-of-the-art classic solver Compass [4] (hence the negative opt gap). This confirms the versatility of the BQ-MDPs.

### Details about new OP and Asym-TSP experiments:

Both problems satisfy the recursivity property which enables our BQ-MDP approach. The BQ-MDPs for Asym-TSP is the same as for TSP. For OP, a construction step is the selection of a node. The BQ-MDP state is (the tail subproblem) OP instance consisting of the remaining nodes and with the maximal distance updated by subtracting the length of the partial solution.

Since an Asym-TSP instance is parametrized by a distance matrix (instead of node coordinates), we had to slightly update our model architecture to accommodate that new input format: we have simply intertwined Graph Convolutional Network layers (1st order model in [3]) with the original attention layers. A diagram of this modified architecture is included in the rebuttal PDF. We kept the model hyperparameters the same as for TSP, CVRP and OP.

### Datasets:

We train our models on 1M instances of ATSP100 (resp. OP100) and the associated high-quality solutions provided by the standard solvers LKH (resp. Compass [4]). The synthetic instances for ATSP are generated following a distribution inspired by (i.e. matching the statistics of) the provided datasets of the recent competition [6].

### Baselines:

For OP, we use the state-of-the-art Compass [4] solver as well as three neural approaches: AM, MDAM and SymNCO [1].
For ATSP, MatNet [5] is the only neural approach. We tested the provided MatNet model (trained on ATSP100) on ATSP100/200 but it could not be applied on instances with 500 and 1000 nodes. The lack of flexibility of the model w.r.t. graph sizes is in fact explained in the paper (Sec 3.3).

[1] Kim, et al, "Sym-NCO: Leveraging symmetricity for neural combinatorial optimization", NeurIPS 2022.
[2] Choo et al, "Simulation-guided Beam Search for Neural Combinatorial Optimization", NeurIPS 2022.
[3] Kipf and Welling, "Semi-Supervised Classification with Graph Convolutional Networks", ICLR 2017.
[4] Kobeaga et al. "An efficient evolutionary algorithm for the orienteering problem", Computers & Operations Research 2018.
[5] Kwon et al, "Matrix Encoding Networks for Neural Combinatorial Optimization",  NeurIPS 2022.
[6] EURO Meets NeurIPS 2022 Vehicle Routing Competition.

---

### Decision · Program_Chairs · 2023-09-21

**Decision:**

Accept (poster)

**Comment:**

All reviewers are supportive of accepting this work, and I agree.

I would suggest the authors incorporate the feedback and suggestions provided by the reviewers in preparation of their camera-ready version.